# GREEDY INFORMATION MAXIMIZATION FOR ACTIVE FEATURE ACQUISITION

## ABSTRACT

Feature selection is commonly used to reduce feature acquisition costs, but the standard approach is to train models with static feature subsets. Here, we consider the *active feature acquisition* problem where the model sequentially queries features based on the presently available information. Active feature acquisition has frequently been addressed using reinforcement learning (RL), but we explore a simpler approach of greedily selecting features based on their conditional mutual information. This approach is theoretically appealing but difficult to implement in practice, so we introduce a learning algorithm based on amortized optimization that recovers the greedy policy when perfectly optimized. We find that the greedy method outperforms both RL-based and static feature selection methods across numerous datasets, which validates our approach as a simple but powerful baseline for this problem.

## 1 INTRODUCTION

Machine learning models require informative inputs to make accurate predictions, but a model's input features can be costly to acquire. In settings where information is gathered sequentially, and particularly when obtaining features requires time or money, it is reasonable to query features adaptively based on the presently available information. We refer to this as *active feature acquisition*[1] (Saar-Tsechansky et al., 2009), and it has been considered by many works in the last decade (Dulac-Arnold et al., 2011; Chen et al., 2015b; Early et al., 2016a; He et al., 2016a; Kachuee et al., 2018).

Feature selection with fixed feature sets (*static* feature selection) has received more attention (see reviews by Li et al. 2017; Cai et al. 2018), but active approaches offer the potential for better performance given a fixed budget. This is easy to see, because selecting the same features for all instances (e.g., all patients visiting a doctor's office) is one possible policy but likely suboptimal in many situations. On the other hand, active approaches are also more challenging because they require both learning a selection policy and making predictions with multiple feature sets.

Prior work has approached active feature acquisition in several ways, but often using reinforcement learning (RL) (Dulac-Arnold et al., 2011; Shim et al., 2018; Kachuee et al., 2018; Janisch et al., 2019; Li & Oliva, 2021). RL is a natural approach for sequential decision-making problems, but current methods are difficult to train and do not reliably outperform static feature selection (Henderson et al., 2018; Erion et al., 2021). Our work therefore explores a simpler approach: sequentially selecting features based on their conditional mutual information (CMI) with the response variable.

The greedy CMI approach is discussed in prior work (Fleuret, 2004; Chen et al., 2015b; Ma et al., 2018), but it remains difficult to implement because it requires perfect knowledge of the joint data distribution. The focus of this work is therefore developing a simple method to approximate the greedy policy. Our main insight is to leverage amortized optimization (Amos, 2022): by developing an optimization-based characterization of the greedy CMI policy, we design an end-to-end learning approach that recovers the policy when it is perfectly optimized.

Our contributions in this work are the following:

---

[1]The problem is also sometimes referred to as *sequential* or *dynamic* feature selection.

1. We develop an optimization-based characterization of the locally optimal selection policy, and we show that this yields greedy selections based on CMI for classification problems. These preliminary results later provide the foundation for our learning approach.

2. We describe a procedure to implement the greedy policy given perfect knowledge of the data distribution. While impractical, this procedure relates our approach to a prior CMI approximation (Ma et al., 2018) and shows that the greedy policy is a special case of existing expected utility frameworks (Saar-Tsechansky et al., 2009; Early et al., 2016a).

3. We develop an amortized optimization approach to approximate the greedy policy using deep learning. Our method permits simple end-to-end learning via stochastic gradient descent, and we prove that our objective recovers the greedy CMI policy when it is perfectly optimized.

We demonstrate the effectiveness of our approach on numerous datasets, and the results show that our method outperforms both RL-based and static feature selection methods. Overall, our work shows that a greedy policy is a simple and powerful method for active feature acquisition.

## 2 PROBLEM FORMULATION

In this section, we introduce notation used throughout the paper and describe the active feature acquisition problem.

### 2.1 NOTATION

Let $\mathbf{x}$ denote a vector of input features and $\mathbf{y}$ a response variable for a supervised learning task. The input consists of $d$ distinct features, or $\mathbf{x} = (\mathbf{x}_1, \ldots, \mathbf{x}_d)$. We use $s \subseteq [d] \equiv \{1, \ldots, d\}$ to denote a subset of indices and $\mathbf{x}_s = \{\mathbf{x}_i : i \in s\}$ a subset of features. Bold symbols $\mathbf{x}, \mathbf{y}$ represent random variables, the symbols $x, y$ are possible values, and $p(\mathbf{x}, \mathbf{y})$ denotes the data distribution.

Our goal is to design a policy that controls which features are selected. The feature selection policy can be viewed as a function $\pi(x_s) \in [d]$, meaning that it receives a subset of features as its input and outputs the next feature index to query. The policy is accompanied by a predictor $f(x_s)$ that can make predictions given the set of available features; for example, if $\mathbf{y}$ is discrete then predictions lie in the probability simplex, or $f(x_s) \in \Delta^{K-1}$ for $K$ classes. The notation $f(x_s \cup x_i)$ represents the prediction given the combined features $x_{s \cup \{i\}}$. The paper initially considers policy and predictor functions that operate on feature subsets because these are simpler to analyze, and Section 4 proposes an implementation using a mask variable $m \in [0, 1]^d$ where the functions operate on $x \odot m$.

### 2.2 ACTIVE FEATURE ACQUISITION

The goal of active feature acquisition is to select features with a minimal budget that achieve maximum predictive accuracy. Access to more features generally makes prediction easier, so the challenge is selecting a small number of informative features. There are multiple formulations for this problem, including non-uniform feature costs and different budgets for each sample (Kachuee et al., 2018), but we focus on the setting with a fixed budget and uniform costs. Our goal is to begin with no features for each data example, sequentially select a feature set $x_s$ such that $|s| = k$ for a fixed $k < d$, and make accurate predictions for the response $y$.

Given a loss function that measures the discrepancy between predictions and labels $\ell(\hat{y}, y)$, a natural scoring criterion is the expected loss after selecting $k$ features. The scoring is applied to a policy-predictor pair $(\pi, f)$, and we define the score for a fixed budget $k$ as follows,

$$v_k(\pi, f) = \mathbb{E}_{p(\mathbf{x}, \mathbf{y})} \big[ \ell \big( f \big( \{\mathbf{x}_{i_t}\}_{t=1}^k \big), \mathbf{y} \big) \big], \tag{1}$$

where feature indices are chosen sequentially for each $(\mathbf{x}, \mathbf{y})$ according to $i_n = \pi(\{\mathbf{x}_{i_t}\}_{t=1}^{n-1})$. Our goal is to minimize $v_k(\pi, f)$, or equivalently to maximize our final predictive accuracy.

One approach is to frame this as a Markov decision process (MDP) and solve it using standard RL techniques, where $\pi$ and $f$ are trained to optimize a reward function based on eq. (1). Indeed, several works have designed such formulations (Dulac-Arnold et al., 2011; Shim et al., 2018; Kachuee et al., 2018; Janisch et al., 2019; Li & Oliva, 2021). Our work instead focuses on a simpler greedy approach, which has the benefits of being simpler to interpret and easier to learn in practice.

## 3 GREEDY INFORMATION MAXIMIZATION

In this section, we define a greedy approach to active feature acquisition and show that it reduces to selecting features based on their CMI with the response variable. We then describe the difficulty of estimating CMI in practice, which motivates our main approach introduced in Section 4.

### 3.1 THE GREEDY POLICY

Greedy algorithms are characterized by making locally optimal choices, and the locally optimal approach here is to sequentially select features that maximize the one-step performance improvement. At any point in the selection process, the greedy approach should select the next feature $\mathbf{x}_i$ that is most helpful given the current features $x_s$, regardless of the feature count $|s|$ and budget $k$.[2]

More precisely, we define the current greedy selection as the index that minimizes the expected prediction loss, similar eq. (1) but with a single additional feature. The expected loss depends on the predictor $f$, but we are interested primarily in the best-case scenario where we use an optimal predictor. Fortunately, the loss-minimizing predictor can be defined independently of the policy $\pi$, assuming that features are selected without knowledge of the remaining features or response variable. We formalize this in the following proposition (see proofs in Appendix A).

**Proposition 1.** *When $\mathbf{y}$ is discrete and $\ell$ is cross-entropy loss, the loss-minimizing predictor $f^*$ for any policy $\pi$ is the Bayes classifier, or $f^*(x_s) = p(\mathbf{y} \mid x_s)$.*

Note that the above property does not hold in problems where selections are based on the entire feature set (Chen et al., 2018; Yoon et al., 2018). Now, assuming that we use the Bayes classifier $f^*$ as a predictor, we define greedy active feature acquisition via a policy $\pi^*$ that minimizes the expected prediction loss at each step, where the expectation is with respect to the unknown variables $(\mathbf{x}_{[d]\setminus s}, \mathbf{y})$ and conditions on the observed features $x_s$.

**Definition 1** (Greedy feature acquisition). *The greedy feature acquisition approach is represented by a policy $\pi^*$ that has the following output given features $x_s$:*

$$\pi^*(x_s) = \arg\min_i \ \mathbb{E}_{\mathbf{y}, \mathbf{x}_i \mid x_s}\big[\ell(f^*(x_s \cup \mathbf{x}_i), \mathbf{y})\big]. \tag{2}$$

Following Definition 1, we find that greedy policy has an intuitive probabilistic interpretation. We focus on the case where $\mathbf{y}$ is discrete and $\ell$ is cross-entropy loss, but our results can also be generalized to regression problems (see Appendix A). In the classification case, we can show the following result.

**Proposition 2.** *Given the features $x_s$, the greedy policy $\pi^*$ selects the feature $\mathbf{x}_i$ having maximum CMI with the response variable, or*

$$\pi^*(x_s) = \arg\max_i \ I(\mathbf{y}; \mathbf{x}_i \mid x_s). \tag{3}$$

According to this result, we can understand the greedy policy as sequentially selecting features that provide maximum information about the response variable. We therefore refer to the policy $\pi^*$ as performing *greedy information maximization*. We can alternatively understand it as performing greedy uncertainty minimization, because maximizing CMI with $\mathbf{y}$ is equivalent to minimizing $\mathbf{y}$'s expected conditional entropy (Cover & Thomas, 2012):

$$\arg\max_i \ I(\mathbf{y}; \mathbf{x}_i \mid x_s) = \arg\min_i \ \mathbb{E}_{\mathbf{x}_i \mid x_s}\big[H(\mathbf{y} \mid x_s, \mathbf{x}_i)\big]. \tag{4}$$

Maximizing the additional information about $\mathbf{y}$ at each step is intuitive and should be an effective approach in many problems. Prior work has discussed CMI for active feature acquisition (Fleuret, 2004; Chen et al., 2015b; Ma et al., 2018), but a reliable implementation has not yet been developed because CMI is challenging to calculate in practice. We next discuss one possible approximation approach that requires perfect knowledge of the data distribution, and we later use the optimization-based results derived here to develop a new learning method (Section 4).

---

[2]Note that this differs from greedily following the Q-network in RL, because the estimated Q-values reflect the reward after multiple selection steps.

## 3.2 AN ITERATIVE IMPLEMENTATION

The greedy policy is trivial to implement if we can directly calculate the CMI $I(\mathbf{y}; \mathbf{x}_i \mid x_s)$, but this is rarely the case in practice. Before introducing our main approach, we first describe a procedure to estimate CMI given access to the response distributions $p(\mathbf{y} \mid \mathbf{x}_s)$ for all $s \subseteq [d]$ and the feature conditional distributions $p(\mathbf{x}_i \mid \mathbf{x}_s)$ for all $s \subseteq [d]$ and $i \in [d]$.

At any point in the selection procedure, given the current features $x_s$, we can iteratively score the features $\mathbf{x}_i$ where $i \notin s$ as follows. First, we can sample multiple values for $\mathbf{x}_i$ from its conditional distribution, or $x_i^j \sim p(\mathbf{x}_i \mid x_s)$ for $j \in [n]$. Next, we can generate Bayes optimal predictions for each sampled value, or $p(\mathbf{y} \mid x_s, x_i^j)$. Finally, we can calculate the mean prediction and the mean KL divergence relative to the mean prediction, which yields the following selection criterion:

$$I_i^n = \frac{1}{n} \sum_{j=1}^{n} D_{\mathrm{KL}}\Big( p(\mathbf{y} \mid x_s, x_i^j) \,\|\, \frac{1}{n} \sum_{l=1}^{n} p(\mathbf{y} \mid x_s, x_i^l) \Big). \tag{5}$$

This per-feature score measures the variability among predictions and captures whether different $\mathbf{x}_i$ values significantly affect $\mathbf{y}$'s conditional distribution. Selecting the next feature according to $\arg\max_i I_i^n$ coincides with the greedy policy due to the following limiting result (see Appendix A):

$$\lim_{n \to \infty} I_i^n = I(\mathbf{y}; \mathbf{x}_i \mid x_s). \tag{6}$$

This procedure thus provides a simple way to identify greedy selections by iteratively scoring features according to $I_i^n$. Similar feature scoring ideas were explored in prior work (Chen et al., 2015a; Early et al., 2016a;b), but the implementation choices in these works prevented them from performing greedy information maximization. Our procedure can be viewed as a special case of the expected utility frameworks from Saar-Tsechansky et al. (2009); Early et al. (2016a), but with a carefully chosen predictor, utility measure, and sampling distribution for missing features.[3]

The iterative procedure is theoretically justified, but it is impractical because we typically lack access to $p(\mathbf{y} \mid \mathbf{x}_s)$ and $p(\mathbf{x}_i \mid \mathbf{x}_s)$. We can instead use learned substitutes for each distribution, e.g., a classifier that approximates $f(x_s) \approx p(\mathbf{y} \mid x_s)$ and a conditional generative model that approximates samples from $p(\mathbf{x}_i \mid \mathbf{x}_s)$. Ma et al. (2018) proposed an approach of this form, but it relies on modified variational autoencoders for generative modeling (Kingma et al., 2015). This approach is limited in practice for several reasons: the difficulty of training reliable generative models, the challenge of working with mixed continuous and categorical features (Ma et al., 2020), and the slow iterative scoring procedure. Section 4 therefore explores a different approach that avoids modeling feature conditional distributions $p(\mathbf{x}_i \mid \mathbf{x}_s)$ by directly learning to output the greedy selections.

## 3.3 GREEDY SUBOPTIMALITY

Before introducing our main approach, we briefly consider the performance gap between the greedy policy and the optimal selection procedure. By definition, the greedy approach cannot achieve better performance than the optimal policy-predictor pair, or

$$\min_{\pi, f} v_k(\pi, f) \leq v_k(\pi^*, f^*). \tag{7}$$

We refer to the difference between the optimal and greedy scores as the *greedy suboptimality gap*, or $G(p, k) = v_k(\pi^*, f^*) - \min_{\pi, f} v_k(\pi, f)$, where the gap depends the data distribution $p(\mathbf{x}, \mathbf{y})$ and feature budget $k$. The greedy suboptimality gap is difficult to assess in practice, but we design several simple scenarios to illustrate that the gap can be large or small.

**Example 1.** *When $k = 1$, the greedy and optimal policies are identical and $G(p, k) = 0$ for all data distributions $p(\mathbf{x}, \mathbf{y})$. The gap becomes non-zero only when selecting $k > 1$ features.*

**Example 2.** *Consider a data distribution $p(\mathbf{x}, \mathbf{y})$ where $\mathbf{x}$ consists of $d$ independent Gaussian variables $\mathbf{x}_i \sim \mathcal{N}(0, \sigma_i^2)$, the variances are ordered as $\sigma_1^2 > \ldots > \sigma_d^2$, and the response is defined as $\mathbf{y} = \sum_{i=1}^{d} \mathbf{x}_i$. If we let $\ell$ be squared error loss, then the greedy policy selects the features in order, or $\mathbf{x}_1, \ldots, \mathbf{x}_k$, and the optimal policy selects the same features in arbitrary order. The gap is therefore $G(p, k) = 0$ for all $1 \leq k \leq d$.*

---

[3]In our approach, we set the utility to the KL divergence with the mean prediction, we assume the Bayes classifier, and we sample features from $p(\mathbf{x}_i \mid x_s)$ rather than $p(\mathbf{x}_i)$.

**Example 3.** *Consider a data distribution $p(\mathbf{x}, \mathbf{y})$ where $\mathbf{x}$ consists of 10 independent Bernoulli variables $\mathbf{x}_i \sim \text{Bern}(0.5)$ and the response is defined as $\mathbf{y} = \text{xor}(\mathbf{x}_1, \mathbf{x}_2)$. Given a budget $k = 2$, the optimal policy selects both $\mathbf{x}_1$ and $\mathbf{x}_2$ and the optimal score is $\min_{f,\pi} v_2(\pi, f) = 0$. In contrast, the greedy policy will fail to discover this because neither $\mathbf{x}_1$ nor $\mathbf{x}_2$ provide any information on their own. The greedy policy is indifferent among the features, and if it fails to select both $\mathbf{x}_1$ and $\mathbf{x}_2$ then it has no information about $\mathbf{y}$ and we have $G(p, 2) = \log 2$ (the entropy of a coin flip).*

The third example illustrates the perils of following a greedy approach: it fails to account for each selection's impact later in the selection procedure. In contrast, a non-greedy policy can make suboptimal selections that yield large improvements in later steps. We find that the greedy approach provides strong performance in practice (Section 6), but understanding the conditions that let us bound the suboptimality gap is an important topic.

In the static feature selection context, numerous works have exploited weak submodularity to prove the effectiveness of greedy algorithms (Das & Kempe, 2011; Elenberg et al., 2017; 2018). The analogous notion of adaptive submodularity (Golovin & Krause, 2011) does not apply to active feature acquisition based on CMI, but Chen et al. (2015b) derived suboptimality bounds based on strict distributional assumptions. More general characterizations are an open topic for future work.

## 4  PROPOSED METHOD

We now introduce our main approach, a practical implementation of the greedy selection policy trained using deep learning.

### 4.1  AN AMORTIZED OPTIMIZATION APPROACH

Instead of iteratively scoring features (Section 3.2), we propose to learn a model that directly outputs the next greedy selection given the current features. Our approach can be learned using a dataset of labeled examples, and it does not require annotations indicating the correct greedy selections. Rather, we cast the optimization-based characterization of the greedy CMI policy (Section 3.1) as an objective function to be optimized by a learnable network, leveraging an approach known as *amortized optimization* (Amos, 2022).

First, because it facilitates gradient-based optimization, we now consider that the policy outputs a distribution over feature indices. With slight abuse of notation, this section lets the policy be a function $\pi(x_s) \in \Delta^{d-1}$, which generalizes the previous definition $\pi(x_s) \in [d]$. Using this stochastic policy, we can now formulate our objective function as follows.

Let the selection policy be parameterized by a neural network $\pi(\mathbf{x}_s; \phi)$ and the predictor by a neural network $f(\mathbf{x}_s; \theta)$. Let $p(\mathbf{s})$ represent a distribution over subsets with $p(s) > 0$ for all $|s| < d$. Then, our objective function $\mathcal{L}(\theta, \phi)$ is defined as

$$\mathcal{L}(\theta, \phi) = \mathbb{E}_{p(\mathbf{x}, \mathbf{y})} \mathbb{E}_{p(\mathbf{s})} \Big[ \mathbb{E}_{i \sim \pi(\mathbf{x}_s; \phi)} \big[ \ell\big( f(\mathbf{x_s} \cup \mathbf{x}_i; \theta), \mathbf{y} \big) \big] \Big]. \tag{8}$$

Intuitively, eq. (8) represents generating a random feature set $\mathbf{x}_s$, sampling a feature index according to $i \sim \pi(\mathbf{x}_s; \phi)$, and then measuring the loss of the prediction $f(\mathbf{x}_s \cup \mathbf{x}_i; \theta)$. Our objective thus optimizes for individual selections and predictions rather than the entire trajectory, which lets us build on earlier results from Section 3. We describe this as an implementation of the greedy approach because it recovers the greedy behavior if it is perfectly optimized. In the classification case, we can show the following result under a mild assumption that there is a unique optimal selection.

**Theorem 1.** *When $\mathbf{y}$ is discrete and $\ell$ is cross-entropy loss, the global optimum of eq. (8) is a predictor that satisfies $f(x_s; \theta^*) = p(\mathbf{y} \mid x_s)$ and a policy $\pi(x_s; \phi^*)$ that puts all probability mass on $i^* = \arg\max_i I(\mathbf{y}; \mathbf{x}_i \mid x_s)$.*

If we relax the assumption of a unique optimal selection, the optimal policy is simply indifferent among the optimal indices. A similar result holds in the regression case, where we can interpret the greedy policy as performing conditional variance minimization.

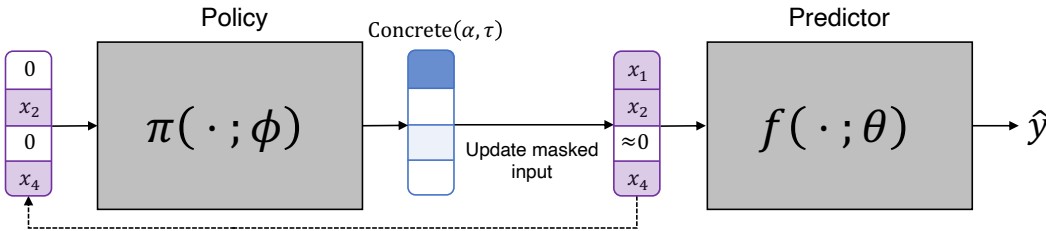

Figure 1: Diagram of our training approach. Left: features are selected by making repeated calls to the policy network using masked inputs. Right: predictions are made after each selection using the predictor network. Solid lines are backpropagated through when performing gradient descent.

**Theorem 2.** *When* $\mathbf{y}$ *is continuous and* $\ell$ *is squared error loss, the global optimum of eq.* (8) *is a predictor that satisfies* $f(x_s; \theta^*) = \mathbb{E}[\mathbf{y} \mid x_s]$ *and a policy* $\pi(x_s; \phi^*)$ *that puts all probability mass on* $i^* = \arg\min_i \mathbb{E}_{\mathbf{x}_i \mid x_s}[\mathrm{Var}(\mathbf{y} \mid x_s, \mathbf{x}_i)]$.

Proofs for these results are in Appendix A. This approach has two key advantages over the iterative procedure from Section 3.2. First, we avoid modeling the conditional distributions $p(\mathbf{x}_i \mid \mathbf{x}_s)$ for all $(s, i)$. Modeling these distributions is a difficult intermediate step, and our approach instead aims to directly output the optimal index. Second, our approach is faster because each index is generated in a single forward pass: selecting $k$ features using the iterative procedure requires $\mathcal{O}(dk)$ scoring steps, but our approach requires only $k$ forward passes through the selector $\pi(\mathbf{x}_s; \phi)$.

Furthermore, compared to a policy learned by RL, the greedy approach is easier to learn. Our training procedure can be understood as a form of reward shaping (Sutton et al., 1998; Randløv & Alstrøm, 1998), where the reward accounts for the loss after each step and provides a strong signal about whether each selection is helpful. In comparison, observing the reward only after selecting $k$ features provides a comparably weak signal to the policy network. RL methods generally face a challenging exploration-exploitation trade-off, but learning the greedy policy is simpler because it only requires finding the locally optimal choice among $\mathcal{O}(d)$ options at each step.

## 4.2 CONTINUOUS RELAXATION

Our objective yields the greedy selection policy when it is perfectly optimized, but $\mathcal{L}(\theta, \phi)$ is difficult to optimize by gradient descent. In particular, gradients are difficult to propagate through the policy network given a sampled index $i \sim \pi(x_s; \phi)$. The REINFORCE trick (Williams, 1992) is one way to get stochastic gradients, but high gradient variance can make it ineffective in many problems. There is a robust literature on reducing gradient variance in this setting (Mnih & Gregor, 2014; Tucker et al., 2017; Grathwohl et al., 2018), but we opt for a simple alternative: the Concrete distribution (Maddison et al., 2016).

An index sampled according to $i \sim \pi(x_s; \phi)$ can be represented by a one-hot vector $m \in \{0, 1\}^d$ indicating the chosen index, and with the Concrete distribution we instead sample an *approximately* one-hot vector in the probability simplex, or $m \in \Delta^{d-1}$. This continuous relaxation lets us calculate gradients via the reparameterization trick (Maddison et al., 2016; Jang et al., 2016). Relaxing the subset $s \subseteq [d]$ to a continuous vector also requires relaxing the policy and predictor functions, so we let these operate on a masked input $x$, or the element-wise product $x \odot m$. To avoid ambiguity about whether features are zero or masked, we can also pass the mask as an input.

Training with the Concrete distribution also requires specifying a temperature parameter $\tau > 0$ to control how discrete its samples are. Previous works have trained with a fixed temperature or annealed it over a pre-determined number of epochs (Chang et al., 2017; Chen et al., 2018; Balın et al., 2019), but we instead train with a sequence of $\tau$ values and perform early stopping at each step. This effectively removes the temperature and number of epochs as important hyperparameters. Our training procedure is summarized in Figure 1 and Algorithm 1.

There are also several optional steps that we found can improve optimization:

- Parameters can be shared between the predictor and policy networks $f(\mathbf{x}; \theta), \pi(\mathbf{x}, \phi)$. This does not complicate their joint optimization, and learning a shared representation in the early network layers can in some cases help the networks optimize faster.

- Rather than training with a random subset distribution $p(\mathbf{s})$, we generate subsets using features selected by the policy $\pi(\mathbf{x}; \phi)$. This allows the models to focus on subsets likely to be encountered at inference time, and it does not affect the globally optimal policy/predictor: gradients are not propagated between selections, so both eq. (8) and this sampling approach treat each feature set as an independent optimization problem, only with different weights (see Appendix D).

- We pre-train the predictor $f(\mathbf{x}; \theta)$ using random subsets before jointly training the policy-predictor pair. This works better than optimizing $\mathcal{L}(\theta, \phi)$ from a random initialization, because a random predictor $f(\mathbf{x}; \theta)$ provides no signal to $\pi(\mathbf{x}; \phi)$ about which features are useful.

## 5 RELATED WORK

Prior work has frequently addressed active feature acquisition using RL. Among these, Dulac-Arnold et al. (2011); Shim et al. (2018); Janisch et al. (2019); Li & Oliva (2021) optimize a reward based on the final prediction accuracy, and Kachuee et al. (2018) use a reward that accounts for prediction uncertainty. RL is a natural approach for sequential decision-making problems, but it can be difficult to learn in practice: RL requires complex architectures and training routines, is slow to converge and is highly sensitive to its initialization (Henderson et al., 2018). As a result, RL does not reliably outperform static feature selection methods in practice (Erion et al., 2021).

Several other approaches include imitation learning (He et al., 2012; 2016a) and iterative feature scoring methods (Melville et al., 2004; Saar-Tsechansky et al., 2009; Chen et al., 2015a; Early et al., 2016b;a). Imitation learning casts active feature acquisition as supervised classification; in comparison, our training approach bypasses the need for an oracle policy. Most existing iterative techniques are greedy methods, like ours, but they use scoring heuristics unrelated to maximizing CMI (Section 3.2). Two iterative feature scoring methods are designed to calculate CMI, but they suffer from practical limitations: Fleuret (2004) assumes binary features, and Ma et al. (2018) relies on difficult-to-train generative models. Our approach is simpler, faster and more flexible, because the selection logic is contained within a learned policy network. Finally, Chen et al. (2015b) study the CMI approach but from a theoretical rather than learning perspective.

Static feature selection is a long-standing problem (Guyon & Elisseeff, 2003; Cai et al., 2018). There are no default approaches for neural networks, but several options include ranking features by local or global importance scores (Breiman, 2001; Shrikumar et al., 2017; Sundararajan et al., 2017; Covert et al., 2020) and differentiable feature selection: several prior works have leveraged continuous relaxations to learn feature selection strategies by gradient descent. For example, Chang et al. (2017); Balın et al. (2019); Yamada et al. (2020); Lindenbaum et al. (2021); Lee et al. (2021) perform static feature selection, and Chen et al. (2018); Jethani et al. (2021) perform instance-wise feature selection given all the features. Our work uses a similar optimization technique but in a different context, and our method learns a selection policy rather than a static selection layer.

## 6 EXPERIMENTS

We now demonstrate the use of our greedy approach on several datasets.[4] We first explore tabular datasets of various sizes, including four medical diagnosis tasks, and we then consider two image classification datasets. Several of the tasks are natural candidates for active feature acquisition, and the remaining ones serve as useful tasks to test the effectiveness of our approach.

As comparisons for our method, we use several active and static baselines. To ensure consistent comparisons, we only use methods applicable to neural networks. As static baselines, we use two methods that rank feature importance based on model accuracy, permutation tests (Breiman, 2001) and SAGE (Covert et al., 2020). We also use two methods that quantify importance on a per-prediction level, DeepLift (Shrikumar et al., 2017) and Integrated Gradients (Sundararajan et al., 2017). We then use a supervised version of Concrete Autoencoders (Balın et al., 2019), a differentiable feature selection method. As active baselines, we use the RL-based Opportunistic Learning (OL) approach (Kachuee et al., 2018), and we follow the authors' implementation with minor modifications (Appendix C). Finally, we use a version of our iterative procedure from Section 3.2: for simplicity, we train a predictor with random feature subsets (similar to our predictor pre-training

---

[4]Link to code removed for anonymity.

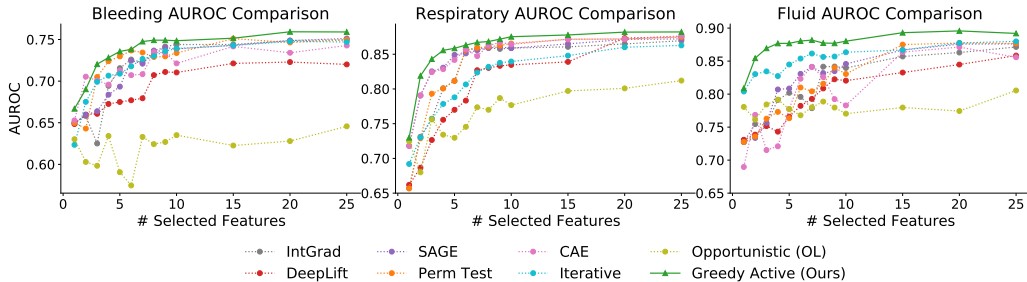

Figure 2: Evaluating the greedy active approach on three medical diagnosis tasks.

in Section 4.2), we sample values from their marginal rather than conditional distributions, and we calculate the scoring criterion in eq. (5) using $n = 128$ samples.

## 6.1 TABULAR DATASETS

We first applied our method to three medical diagnosis tasks derived from an emergency medicine setting. The tasks involve predicting a patient's bleeding risk via a low fibrinogen concentration (bleeding), whether the patient requires endotracheal intubation for respiratory support (respiratory), and whether the patient will be responsive to fluid resuscitation (fluid). Appendix B provides more dataset details. In each scenario, gathering all possible inputs is challenging due to time and resource constraints, thus making active feature acquisition a natural solution.

Figure 2 shows the results of applying each method with various feature budgets. We use fully connected architectures for all methods, and we use dropout to reduce overfitting (Srivastava et al., 2014). The classification accuracy is measured via AUROC, and the greedy method achieves the best results at nearly all feature budgets for all three tasks. Among the baselines, several static methods are sometimes close, but OL provides unstable results and the iterative method is only competitive on the fluid task. The greedy method's advantage is largest when selecting a small number of features, and it becomes narrower once the accuracy saturates.

Next, we conducted experiments using three publicly available tabular datasets: spam classification (Dua & Graff, 2017), particle identification (MiniBooNE) (Roe et al., 2005) and diabetes diagnosis (Miller, 1973). The diabetes task is a natural application for active feature acquisition and was used in prior work (Kachuee et al., 2018). We again tested various feature numbers, and Figure 5 shows plots of the AUROC for each budget; on these tasks, the greedy method is most accurate for all numbers of features. Table 1 summarizes the results via the mean AUROC across $k = 1, \ldots, 10$ features, emphasizing the benefits of the greedy method across all six datasets. Appendix E also shows plots demonstrating the variability of selections across samples for each dataset.

The results on these datasets reveal that active methods can often be outperformed by static methods. Interestingly, this point was not highlighted in prior work because strong static selection methods were not always used (Kachuee et al., 2018; Janisch et al., 2019). Active methods are in principle capable of performing better, so the sub-par results from OL and the iterative method underscore the difficulty of learning both a selection policy and a prediction function for multiple feature sets. In our experiments across these six datasets, ours is the only active method to do both successfully. The greedy approach trains more easily for the reasons described in Section 4, and we suspect that OL has difficulty optimizing for larger feature budgets in particular due to its delayed rewards.

Table 1: AUROC averaged across budgets of 1-10 features ($\pm$ one standard deviation).

|  |  | Spam | MiniBooNE | Diabetes | Bleeding | Respiratory | Fluid |
|---|---|---|---|---|---|---|---|
| Static | IntGrad | $88.08 \pm 0.36$ | $86.77 \pm 0.86$ | $88.99 \pm 0.34$ | $69.27 \pm 1.18$ | $80.42 \pm 0.18$ | $78.31 \pm 0.82$ |
|  | DeepLift | $89.02 \pm 1.51$ | $86.33 \pm 0.20$ | $94.89 \pm 0.21$ | $67.52 \pm 0.27$ | $77.66 \pm 0.31$ | $77.37 \pm 0.15$ |
|  | SAGE | $90.45 \pm 0.65$ | $91.57 \pm 0.10$ | $95.46 \pm 0.03$ | $70.31 \pm 0.38$ | $82.20 \pm 0.37$ | $79.60 \pm 0.58$ |
|  | Perm Test | $89.83 \pm 0.34$ | $92.18 \pm 0.15$ | $95.50 \pm 0.02$ | $68.29 \pm 0.88$ | $80.98 \pm 0.27$ | $79.24 \pm 0.71$ |
|  | CAE | $92.46 \pm 0.29$ | $93.04 \pm 0.16$ | $95.88 \pm 0.11$ | $70.03 \pm 0.37$ | $82.78 \pm 0.33$ | $76.39 \pm 1.60$ |
| Active | Iterative | $87.00 \pm 1.40$ | $92.17$ | $95.61$ | $70.63$ | $78.85$ | $84.33$ |
|  | Opportunistic (OL) | $85.93 \pm 0.22$ | $69.22 \pm 0.73$ | $83.06 \pm 0.93$ | $60.60 \pm 0.62$ | $74.44 \pm 0.48$ | $78.12 \pm 0.35$ |
|  | Greedy Active (Ours) | $\mathbf{94.22 \pm 0.20}$ | $\mathbf{94.39}$ | $\mathbf{96.04}$ | $\mathbf{72.83 \pm 0.31}$ | $\mathbf{84.52}$ | $\mathbf{86.00 \pm 0.31}$ |

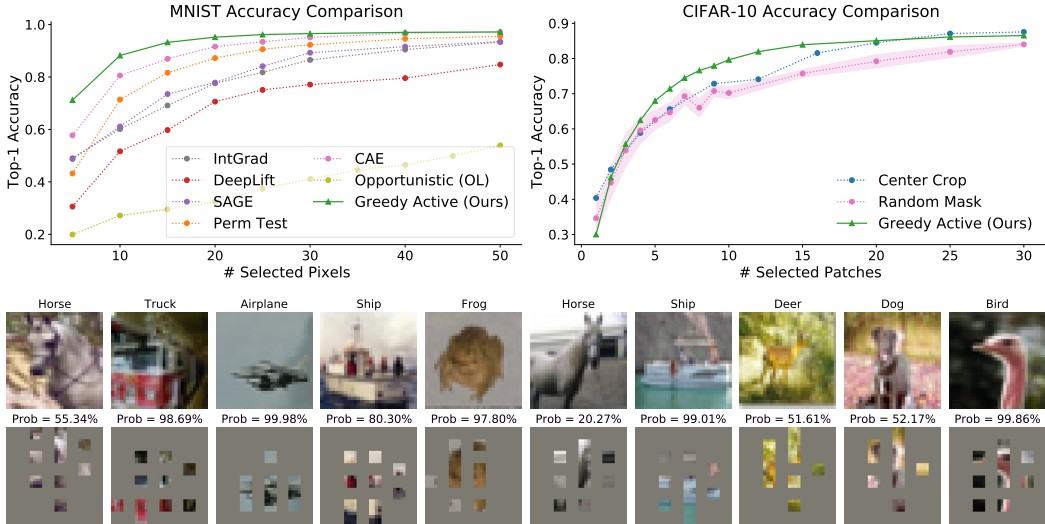

Figure 3: Greedy active feature acquisition for image classification. Top left: accuracy comparison on MNIST. Top right: accuracy comparison on CIFAR-10. Bottom: example selections and predictions for the greedy approach given 10 out of 64 patches for CIFAR-10 images.

## 6.2 IMAGE CLASSIFICATION DATASETS

Next, we considered two standard image classification datasets: MNIST (LeCun et al., 1998) and CIFAR-10 (Krizhevsky et al., 2009). Our goal is to begin with a blank image, sequentially reveal multiple pixels or patches, and ultimately make a classification using a small portion of the image. This is not an obvious use case for active feature acquisition, but it represents a challenging problem for our method, and similar tasks have been considered in several earlier works (Karayev et al., 2012; Mnih et al., 2014; Early et al., 2016a; Janisch et al., 2019).

For MNIST, we use fully connected architectures for both the policy and predictor, and we treat pixels as individual features (where $d = 784$). For CIFAR-10, we use a shared ResNet backbone (He et al., 2016b) for the policy and predictor networks, and each network uses its own output head. The $32 \times 32$ images are coarsened into $d = 64$ patches of size $4 \times 4$, so the selector head generates logits corresponding to each patch, and the predictor head generates probabilities for each class.

Figure 3 shows our method's accuracy for different feature budgets. For MNIST, we use the previous baselines but exclude the iterative method due to its computational cost. We observe a large performance gap, particularly when making a small number of selections: the greedy method reaches nearly 90% accuracy with just 10 pixels, which is nearly 10% higher than the best baseline and considerably higher than prior work (Balın et al., 2019; Yamada et al., 2020; Covert et al., 2020). OL yields the worst results, and it trains slowly, likely due to the large number of states. For CIFAR-10, we use two simple baselines: center crops and random masks of various sizes. For the random baseline, we plot the mean and standard deviation calculated across ten random masks. Our greedy approach is slightly less accurate with 1-2 patches, but it reaches significantly higher accuracy when using 5-20 patches. Figure 3 also shows qualitative examples of our method's predictions after selecting 10 out of 64 patches, and Appendix E shows similar plots with different numbers of patches.

## 7 CONCLUSION

In this work, we developed a greedy approach to active feature acquisition that selects features based on their CMI with the response variable. We then proposed an approach to learn the greedy policy using amortized optimization, and we conducted experiments that show our method outperforms a variety of existing feature selection methods. Future work on this topic may include incorporating non-uniform features costs, automatically determining the feature budget on a per-sample basis, and further characterizing the greedy suboptimality gap. Some progress has been made in analyzing the suboptimality gap in the active feature acquisition setting (Chen et al., 2015b), but more general characterizations remain an open topic for future work.

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

# A PROOFS

In this section, we re-state and prove our main theoretical results. We begin with our proposition regarding the optimal predictor $f^*$ for an arbitrary policy $\pi$.

**Proposition 1.** *When $\mathbf{y}$ is discrete and $\ell$ is cross-entropy loss, the loss-minimizing predictor $f^*$ for any policy $\pi$ is the Bayes classifier, or $f^*(x_s) = p(\mathbf{y} \mid x_s)$.*

*Proof.* The optimal prediction given $x_s$ is defined by minimizing the expected loss, where the expectation is with respect to $\mathbf{y}$ and we must condition on the observed values $x_s$. Because features are selected sequentially by $\pi$ with no knowledge of the non-selected values, there is no other information to condition on. We can therefore derive the optimal prediction as follows:

$$
\begin{aligned}
f^*(x_s) &= \arg\min_{\hat{y}} \; \mathbb{E}_{\mathbf{y}|x_s}\big[\ell(\hat{y}, \mathbf{y})\big] \\
&= \arg\min_{\hat{y}} \; \sum_{i \in \mathcal{Y}} p(\mathbf{y} = i \mid x_s) \log \hat{y}_i \\
&= \arg\min_{\hat{y}} \; D_{\mathrm{KL}}\big(p(\mathbf{y} \mid x_s) \,\|\, \hat{y}\big) + H(\mathbf{y} \mid x_s) \\
&= p(\mathbf{y} \mid x_s).
\end{aligned}
$$

In the case of a continuous response $\mathbf{y} \in \mathbb{R}$ with squared error loss, we have a similar result involving the response's conditional expectation:

$$
\begin{aligned}
f^*(x_s) &= \arg\min_{\hat{y}} \; \mathbb{E}_{\mathbf{y}|x_s}\big[(\hat{y} - \mathbf{y})^2\big] \\
&= \arg\min_{\hat{y}} \; \mathbb{E}_{\mathbf{y}|x_s}\big[(\hat{y} - \mathbb{E}[\mathbf{y} \mid x_s])^2\big] + \mathrm{Var}(\mathbf{y} \mid x_s) \\
&= \mathbb{E}[\mathbf{y} \mid x_s].
\end{aligned}
$$

$\square$

**Proposition 2.** *Given the features $x_s$, the greedy policy $\pi^*$ selects the feature $\mathbf{x}_i$ having maximum conditional mutual information with the response variable, or*

$$
\pi^*(x_s) = \arg\max_i \; I(\mathbf{y}; \mathbf{x}_i \mid x_s). \tag{9}
$$

*Proof.* Consider the expected loss from observing an additional feature $\mathbf{x}_i$ given the values $x_s$. The only available information is $x_s$, so our expected loss is with respect to the conditional distribution $\mathbf{y}, \mathbf{x}_i \mid x_s$. We thus have the following expected loss:

$$
\begin{aligned}
\mathbb{E}_{\mathbf{y}, \mathbf{x}_i | x_s}\big[\ell(f^*(x_s \cup \mathbf{x}_i), \mathbf{y})\big] &= \mathbb{E}_{\mathbf{y}, \mathbf{x}_i | x_s}\big[\ell(p(\mathbf{y} \mid x_s, \mathbf{x}_i), \mathbf{y})\big] \\
&= \mathbb{E}_{\mathbf{x}_i | x_s}\Big[\mathbb{E}_{\mathbf{y}|x_s, \mathbf{x}_i}[\ell(p(\mathbf{y} \mid x_s, \mathbf{x}_i), \mathbf{y})]\Big] \\
&= \mathbb{E}_{\mathbf{x}_i | x_s}\big[H(\mathbf{y} \mid x_s, \mathbf{x}_i)\big] \\
&= H(\mathbf{y} \mid x_s) - I(\mathbf{y}; \mathbf{x}_i \mid x_s).
\end{aligned}
$$

Note that $H(\mathbf{y} \mid x_s)$ is a constant that does not depend on $i$. When identifying the index that yields the minimum expected loss, we thus have the following result:

$$
\arg\min_i \; \mathbb{E}_{\mathbf{y}, \mathbf{x}_i | x_s}\big[\ell(f^*(x_s \cup \mathbf{x}_i), \mathbf{y})\big] = \arg\max_i \; I(\mathbf{y}; \mathbf{x}_i \mid x_s).
$$

In the case of a continuous response with squared error loss and an optimal predictor given by $f^*(x_s) = \mathbb{E}[\mathbf{y} \mid x_s]$, we have a similar result:

$$
\begin{aligned}
\mathbb{E}_{\mathbf{y}, \mathbf{x}_i \mid x_s}\left[(f^*(x_s \cup \mathbf{x}_i) - \mathbf{y})^2\right] &= \mathbb{E}_{\mathbf{y}, \mathbf{x}_i \mid x_s}\left[(\mathbb{E}[\mathbf{y} \mid x_s, \mathbf{x}_i] - \mathbf{y})^2\right] \\
&= \mathbb{E}_{\mathbf{x}_i \mid x_s}\left[\mathbb{E}_{\mathbf{y} \mid x_s, \mathbf{x}_i}[(\mathbb{E}[\mathbf{y} \mid x_s, \mathbf{x}_i] - \mathbf{y})^2]\right] \\
&= \mathbb{E}_{\mathbf{x}_i \mid x_s}[\mathrm{Var}(\mathbf{y} \mid x_s, \mathbf{x}_i)].
\end{aligned}
$$

When selecting the index with minimum expected loss, our selection is thus the index that yields the lowest expected conditional variance:

$$
\arg\min_i \ \mathbb{E}_{\mathbf{x}_i \mid x_s}[\mathrm{Var}(\mathbf{y} \mid x_s, \mathbf{x}_i)].
$$

$\square$

We also prove the limiting result presented in eq. (6), which states that $I_i^n \to I(\mathbf{y}; \mathbf{x}_i \mid x_s)$.

*Proof.* Conditional mutual information $I(\mathbf{y}; \mathbf{x}_i \mid x_s)$ is defined as follows (Cover & Thomas, 2012):

$$
I(\mathbf{y}; \mathbf{x}_i \mid x_s) = \mathbb{E}_{\mathbf{y}, \mathbf{x}_i \mid x_s}\left[\log \frac{p(\mathbf{y}, \mathbf{x}_i \mid x_s)}{p(\mathbf{x}_i \mid x_s)p(\mathbf{y} \mid x_s)}\right].
$$

Rearranging terms, we can also write it as an expected KL divergence:

$$
\begin{aligned}
I(\mathbf{y}; \mathbf{x}_i \mid x_s) &= \mathbb{E}_{\mathbf{x}_i \mid x_s}\mathbb{E}_{\mathbf{y} \mid x_s, \mathbf{x}_i}\left[\log \frac{p(\mathbf{y}, \mathbf{x}_i \mid x_s)}{p(\mathbf{x}_i \mid x_s)p(\mathbf{y} \mid x_s)}\right] \\
&= \mathbb{E}_{\mathbf{x}_i \mid x_s}\mathbb{E}_{\mathbf{y} \mid x_s, \mathbf{x}_i}\left[\log \frac{p(\mathbf{y} \mid x_s, \mathbf{x}_i)}{p(\mathbf{y} \mid x_s)}\right] \\
&= \mathbb{E}_{\mathbf{x}_i \mid x_s}\left[D_{\mathrm{KL}}\big(p(\mathbf{y} \mid x_s, \mathbf{x}_i) \,||\, p(\mathbf{y} \mid x_s)\big)\right]
\end{aligned}
$$

Now, when we sample multiple values $x_i^1, \ldots, x_i^n \sim p(\mathbf{x}_i \mid x_s)$ and make predictions using the Bayes classifier, we have the following mean prediction as $n$ becomes large:

$$
\lim_{n \to \infty} \frac{1}{n}\sum_{j=1}^n p(\mathbf{y} \mid x_s, x_i^j) = \mathbb{E}_{\mathbf{x}_i \mid x_s}\left[p(\mathbf{y} \mid x_s, \mathbf{x}_i)\right] = p(\mathbf{y} \mid x_s).
$$

Calculating the mean KL divergence across each of the predictions, we arrive at the following result:

$$
\lim_{n \to \infty} I_i^n = \mathbb{E}_{\mathbf{x}_i \mid x_s}\left[D_{\mathrm{KL}}\big(p(\mathbf{y} \mid x_s, \mathbf{x}_i) \,||\, p(\mathbf{y} \mid x_s)\big)\right] = I(\mathbf{y}; \mathbf{x}_i \mid x_s).
$$

$\square$

**Theorem 1.** *When $\mathbf{y}$ is discrete and $\ell$ is cross-entropy loss, the global optimum of eq. (8) is a predictor that satisfies $f(x_s; \theta^*) = p(\mathbf{y} \mid x_s)$ and a policy $\pi(x_s; \phi^*)$ that puts all probability mass on $i^* = \arg\max_i I(\mathbf{y}; \mathbf{x}_i \mid x_s)$.*

*Proof.* We first consider the predictor network $f(\mathbf{x}_s; \theta)$. When the predictor is given the feature values $x_s$, it means that one index $i \in s$ was chosen by the policy according to $\pi(x_{s \setminus i}; \phi)$ and the remaining indices $s \setminus i$ were sampled from $p(\mathbf{s})$. Because $\mathbf{s}$ is sampled independently from $(\mathbf{x}, \mathbf{y})$, and because $\pi(x_{s \setminus i}; \phi)$ is not given access to $(\mathbf{x}_{[d] \setminus s}, \mathbf{x}_i, \mathbf{y})$, the predictor's expected loss must be

considered with respect to the distribution $\mathbf{y} \mid x_s$. The globally optimal predictor $f(x_s; \theta^*)$ is thus defined as follows, regardless of the selection policy $\pi(x_s; \phi)$ and which index $i$ was selected last:

$$f(x_s; \theta^*) = \arg\min_{\hat{y}} \; \mathbb{E}_{\mathbf{y} \mid x_s}\big[\ell(\hat{y}, \mathbf{y})\big] = p(\mathbf{y} \mid x_s).$$

The above result follows from our proof for Proposition 1. Now, given the optimal predictor $f(x_s; \theta^*)$, we can define the globally optimal policy by minimizing the expected loss for a fixed input $x_s$. Denoting the probability mass placed on each index $i \in [d]$ as $\pi_i(x_s; \phi)$, where $\pi(x_s; \phi) \in \Delta^{d-1}$, the expected loss is the following:

$$\mathbb{E}_{i \sim \pi(x_s;\phi)} \mathbb{E}_{\mathbf{y}, \mathbf{x}_i \mid x_s}\big[\ell(f(x_s \cup \mathbf{x}_i; \theta^*), \mathbf{y})\big] = \sum_{i \in [d]} \pi_i(x_s; \phi) \mathbb{E}_{\mathbf{y}, \mathbf{x}_i \mid x_s}\big[\ell(f(x_s \cup \mathbf{x}_i; \theta^*), \mathbf{y})\big]$$

$$= \sum_{i \in [d]} \pi_i(x_s; \phi) \mathbb{E}_{\mathbf{x}_i \mid x_s}[H(\mathbf{y} \mid x_s, \mathbf{x}_i)].$$

The above result follows from our proof for Proposition 2. If there exists a single index $i^* \in [d]$ that yields the lowest expected conditional entropy, or

$$\mathbb{E}_{\mathbf{x}_{i^*} \mid x_s}[H(\mathbf{y} \mid x_s, \mathbf{x}_{i^*})] < \mathbb{E}_{\mathbf{x}_i \mid x_s}[H(\mathbf{y} \mid x_s, \mathbf{x}_i)] \quad \forall i \neq i^*,$$

then the optimal predictor must put all its probability mass on $i^*$, or $\pi_{i^*}(x_s; \phi^*) = 1$. Note that the corresponding feature $\mathbf{x}_{i^*}$ has maximum conditional mutual information with $\mathbf{y}$, because we have

$$I(\mathbf{y}; \mathbf{x}_{i^*} \mid x_s) = \underbrace{H(\mathbf{y} \mid x_s)}_{\text{Constant}} - \mathbb{E}_{\mathbf{x}_{i^*} \mid x_s}[H(\mathbf{y} \mid x_s, \mathbf{x}_{i^*})].$$

To summarize, we derived the global optimum to our objective $\mathcal{L}(\theta, \phi)$ by first considering the optimal predictor $f(\mathbf{x}_s; \theta^*)$, and then the optimal policy $\pi(\mathbf{x}_s; \phi^*)$ assuming the optimal predictor. $\square$

**Theorem 2.** *When $\mathbf{y}$ is continuous and $\ell$ is squared error loss, the global optimum of eq. (8) is a predictor that satisfies $f(x_s; \theta^*) = \mathbb{E}[\mathbf{y} \mid x_s]$ and a policy $\pi(x_s; \phi^*)$ that puts all probability mass on $i^* = \arg\min_i \mathbb{E}_{\mathbf{x}_i \mid x_s}[\mathrm{Var}(\mathbf{y} \mid x_s, \mathbf{x}_i)]$.*

*Proof.* Our proof follows the same logic as our proof for Theorem 1. For the optimal predictor given an arbitrary policy, we have:

$$f(x_s; \theta^*) = \arg\min_{\hat{y}} \; \mathbb{E}_{\mathbf{y} \mid x_s}\big[(\hat{y} - \mathbf{y})^2\big] = \mathbb{E}[\mathbf{y} \mid x_s].$$

Then, for the policy's expected loss, we have:

$$\mathbb{E}_{i \sim \pi(x_s;\phi)} \mathbb{E}_{\mathbf{y}, \mathbf{x}_i \mid x_s}\big[\big(f(x_s \cup \mathbf{x}_i; \theta^*) - \mathbf{y}\big)^2\big] = \sum_{i \in [d]} \pi_i(x_s; \phi) \mathbb{E}_{\mathbf{x}_i \mid x_s}[\mathrm{Var}(\mathbf{y} \mid x_s, \mathbf{x}_i)].$$

If there exists an index $i^* \in [d]$ that yields the lowest expected conditional variance, then the optimal policy must put all its probability mass on $i^*$, or $\pi_{i^*}(x_s; \phi^*) = 1$. $\square$

# B   DATASETS

The datasets used in our experiments are summarized in Table 2. Three of the tabular datasets and the two image classification datasets are publicly available, and the three emergency medicine tasks were privately curated from the Harborview Medical Center Trauma Registry.

Table 2: Summary of datasets used in our experiments.

| Dataset | # Features | # Feature Groups | # Classes | # Samples |
|---|---|---|---|---|
| Fluid | 224 | 162 | 2 | 2,850 |
| Respiratory | 112 | 37 | 2 | 65,515 |
| Bleeding | 121 | 46 | 2 | 6,496 |
| Spam | 58 | – | 2 | 4,601 |
| MiniBooNE | 51 | – | 2 | 130,064 |
| Diabetes | 45 | – | 3 | 92,062 |
| MNIST | 784 | – | 10 | 60,000 |
| CIFAR-10 | 1,024 | 64 | 10 | 60,000 |

## B.1   MINIBOONE AND SPAM CLASSIFICATION

The spam dataset includes features extracted from e-mail messages to predict whether or not a message is spam. Three features describes the shortest, average, and total length of letters in the message, and the remaining 54 features describe the frequency with which certain key words are used. The MiniBooNE particle identification dataset involves distinguishing electron neutrinos from muon neutrinos based on various continuous features (Roe et al., 2005). Both datasets were obtained from the UCI repository (Dua & Graff, 2017).

## B.2   DIABETES CLASSIFICATION

This dataset is obtained from from the National Health and Nutrition Examination Survey (NHANES) (NHA, 2018), an ongoing survey designed to assess the well-being of adults and children in the United States. We used a version of the data pre-processed by Kachuee et al. (2018; 2019) that includes data collected from 1999 through 2016. The input features include demographic information (age, gender, ethnicity, etc.), lab results (total cholesterol, triglyceride, etc.), examination data (weight, height, etc.), and questionnaire answers (smoking, alcohol, sleep habits, etc.). An expert was also asked to suggest costs for each feature based on the financial burden, patient privacy, and patient inconvenience, but we assume uniform feature costs in our experiments for simplicity. Finally, the fasting glucose values were used to define three classes based on standard threshold values: normal, pre-diabetes, and diabetes.

## B.3   IMAGE CLASSIFICATION DATASETS

The MNIST and CIFAR-10 datasets were downloaded using PyTorch (Paszke et al., 2017). We used the standard train-test splits, and we split the train set to get a validation set with the same size as the test set.

## B.4   EMERGENCY MEDICINE DATASETS

The emergency medicine data used in this study was gathered over a 13-year period (2007-2020) and encompasses 14,463 emergency department admissions. We excluded patients under the age of 18, and we curated 3 clinical cohorts commonly seen in pre-hospitalization settings. These include 1) pre-hospital fluid resuscitation, 2) emergency department respiratory support, and 3) bleeding after injury. These datasets are not publicly available due to patient privacy concerns.

**Pre-hospital fluid resuscitation** We selected 224 variables that were available in the pre-hospital setting, including dispatch information (injury date, time, cause, and location), demographic information (age, sex), and pre-hospital vital signs (blood pressure, heart rate, respiratory rate). The outcome was each patient's response to fluid resuscitation, following the Advanced Trauma Life Support (ATLS) definition (Subcommittee et al., 2013).

**Emergency department respiratory support** In this cohort, our goal is to predict which patients require respiratory support upon arrival in the emergency department. Similar to the previous dataset, we selected 112 pre-hospital clinical features including dispatch information (injury date, time, cause, and location), demographic information (age, sex), and pre-hospital vital signs (blood pressure, heart rate, respiratory rate). The outcome is defined based on whether a patient received respiratory support, including both invasive (intubation) and non-invasive (BiPap) approaches.

**Bleeding** In this cohort, we only included patients whose fibrinogen levels were measured, as this provides an indicator for bleeding or fibrinolysis (Mosesson, 2005). Like the previous datasets, demographic information, dispatch information, and pre-hospital observations were used as input features. The outcome, based on experts' opinion, was defined by whether an individual's fibrinogen level is below 200 mg/dL, which represents higher risk of bleeding after injury.

## C  BASELINES

This section provides more details on the baseline methods used in our experiments (Section 6).

### C.1  GLOBAL FEATURE IMPORTANCE METHODS

Two of our static feature selection baselines, permutation tests and SAGE, are known as *global feature importance methods* because they rank features based on their role in improving model accuracy (Covert et al., 2021). In our experiments, we run each method using a single classifier trained on the entire dataset, and we then select the top $k$ features depending on the budget.

When running the permutation test, we calculate the validation AUROC while replacing values in the corresponding feature column with random draws from the training set. When running SAGE, we use the authors' implementation with automatic convergence detection (Covert et al., 2020). For the six tabular datasets, we averaged across 128 sampled values for the held-out features, and for MNIST we used a zeros baseline to achieve faster convergence.

### C.2  LOCAL FEATURE IMPORTANCE METHODS

Two of our static feature selection baselines, DeepLift and Integrated Gradients, are known as *local feature importance methods* because they rank features based on their importance to a single prediction. In our experiments, we generate feature importance scores for the true class using all examples in the validation set. We then select the top $k$ features based on their mean absolute importance. We use a mean baseline for Integrated Gradients (Sundararajan et al., 2017), and both methods were run using the Captum package (Kokhlikyan et al., 2020).

### C.3  OPPORTUNISTIC LEARNING

Kachuee et al. (2018) proposed Opportunistic Learning (OL), an approach to solve active feature acquisition using RL. The model consists of two networks analogous to our policy and predictor: a Q-network that estimates the value associated with each action, where actions correspond to features, and a P-network responsible for making predictions. When using OL, we use the same architectures as our approach, and OL shares network parameters between the P- and Q-networks.

Kachuee et al. (2018) introduce a utility function for their reward, shown in eq. (10), which calculates the difference in prediction uncertainty as approximated by MC dropout (Gal & Ghahramani, 2016). The reward also accounts for feature costs, but we set all feature costs to $c_i = 1$.

$$r_i = \frac{||Cert(x_s) - Cert(x_s \cup x_i)||}{c_i} \tag{10}$$

To provide a fair comparison with the remaining methods, we made several modifications to the authors' implementation. These include 1) preventing the prediction action until the pre-specified budget is met, 2) setting all feature costs to be identical, and 3) supporting pre-defined feature groups as described in Appendix D.3. When training, we update the P-, Q-, and target Q-networks every $1 + \frac{d}{100}$ experiences, where $d$ is the number of features in a dataset. In addition, the replay buffer is set to store the $1000d$ most recent experiences, and the random exploration probability is decayed so that it eventually reaches a value of 0.1.

## D TRAINING APPROACH AND HYPERPARAMETERS

This section provides more details on our training approach and selected hyperparameters.

### D.1 TRAINING PSEUDOCODE

Algorithm 1 summarizes our training approach. Briefly, we select features by drawing a Concrete sample using policy network's logits, we calculate the loss based on the subsequent prediction, and we then update the mask for the next step using a discrete sample from the policy's distribution. We implemented this approach using PyTorch (Paszke et al., 2017) and PyTorch Lightning[5].

---

**Algorithm 1:** Training pseudocode

**Input:** Data distribution $p(\mathbf{x}, \mathbf{y})$, budget $k > 0$, learning rate $\gamma > 0$, temperature $\tau > 0$
**Output:** Predictor model $f(\mathbf{x}; \theta)$, selector model $\pi(\mathbf{x}; \phi)$
initialize $f(\mathbf{x}; \theta), \pi(\mathbf{x}; \phi)$
**while** *not converged* **do**
    sample $x, y \sim p(\mathbf{x}, \mathbf{y})$
    initialize $\mathcal{L} = 0, m = [0, \ldots, 0]$
    **for** $j = 1$ **to** $k$ **do**
        calculate logits $\alpha = \pi(x \odot m; \phi)$, sample $G_i \sim$ Gumbel for $i \in [d]$
        set $\tilde{m} = \max\big(m, \text{softmax}(G + \alpha, \tau)\big)$ // update with Concrete
        set $m = \max\big(m, \text{softmax}(G + \alpha, 0)\big)$ // update with one-hot
        update $\mathcal{L} \leftarrow \mathcal{L} + \ell\big(f(x \odot \tilde{m}; \theta), y\big)$
    **end**
    update $\theta \leftarrow \theta - \gamma \nabla_\theta \mathcal{L}, \;\; \phi \leftarrow \phi - \gamma \nabla_\phi \mathcal{L}$
**end**
**return** $f(\mathbf{x}; \theta), \pi(\mathbf{x}; \phi)$

---

One notable difference between Algorithm 1 and our objective $\mathcal{L}(\theta, \phi)$ in the main text is the use of the policy $\pi(\mathbf{x}; \phi)$ for generating feature subsets. This differs from eq. (8), which generates feature subsets using a subset distribution $p(\mathbf{s})$. The key shared factor for both approaches is that there are separate optimization problems over each feature subset that can be viewed independently. For each feature set $x_s$, the problem incorporates both the policy and predictor and can be written as follows:

$$\mathbb{E}_{i \sim \pi(\mathbf{x_s}; \phi)}\big[\ell\big(f(\mathbf{x_s} \cup \mathbf{x}_i; \theta), \mathbf{y}\big)\big]. \tag{11}$$

The problems for each subset do not interact in the sense that during optimization, the selection given $x_s$ is based only on the immediate change in the loss; gradients are not propagated through multiple selections, as they would be for an RL-based solution. In solving these multiple problems, the difference is simply that eq. (8) weights them according to $p(\mathbf{s})$, whereas Algorithm 1 weights them according to the current policy $\pi(\mathbf{x}, \phi)$.

---

[5] https://www.pytorchlightning.ai

### D.2 HYPERPARAMETERS

Our experiments with the six tabular datasets all used fully connected architectures with dropout in all layers (Srivastava et al., 2014). The dropout probability is set to 0.3, the networks have two hidden layers of width 128, and we performed early stopping using the validation loss. For our method, the predictor and selector were separate networks with identical architectures. When training models with the features selected by static methods, we reported results using the best model from multiple training runs based on the validation loss. We did not perform any additional hyperparameter tuning due to the large number of models being trained.

For MNIST, we used fully connected architectures with two layers of width 512 and the dropout probability set to 0.3. Again, our method used separate networks with identical architectures. For CIFAR-10, we used a shared ResNet backbone (He et al., 2016b) consisting of several residually connected convolutional layers. The classification head consists of global average pooling and a linear layer, and the selection head consisted of a transposed convolution layer followed by a $1 \times 1$ convolution, which output a grid of logits with size $8 \times 8$. Our CIFAR-10 networks are trained using random crops and random horizontal flips as augmentations.

### D.3 FEATURE GROUPING

All of the methods used in our experiments were designed to select individual features, but this is undesirable when using categorical features with one-hot encodings. Each of our three emergency medicine tasks involve such features, so we extended each method to support feature grouping.

SAGE and permutation tests are trivial to extend to feature groups: we simply removed groups of features rather than individual ones. For DeepLift and Integrated Gradients, we used the summed importance within each group, which preserves each method's additivity property. For the differentiable method based on Concrete Autoencoders (Balın et al., 2019), we implemented a generalized version of the selection layer that operates on feature groups. We also extended OL to operate on feature groups by having the actions map to groups rather than individual features.

Finally, for our method, we parameterized the policy network $\pi(\mathbf{x}; \phi)$ so that the number of outputs is the number of groups $g$ rather than the total number of features $d$ (where $g \leq d$). When applying masking, we first generate a binary mask $m \in [0, 1]^g$, and we then project the mask into $[0, 1]^d$ using a binary groups matrix $M \in \{0, 1\}^{d \times g}$, where $M_{ij} = 1$ if feature $i$ is in group $j$ and $M_{ij} = 0$ otherwise. Thus, our masked input vector is given by $x \odot (Mm)$.

## E ADDITIONAL RESULTS

This section provides several additional experimental results. First, Figure 4 shows the same results as Figure 2 but larger for improved visibility. Next, Figure 5 shows the AUROC for each feature selection method under different feature budgets for the three publicly available tabular datasets. Here, our greedy active approach achieves the best results for all feature budgets on all three tasks. Notably, our two active baselines are often outperformed by static methods, and OL remains unstable.

Next, Figure 6 though Figure 11 display the feature selection frequency for each of the tabular datasets when using the greedy active method. The heatmaps in each plot show the portion of the time that a feature (or feature group) is selected under a specific feature budget. These plots reveal that our method is indeed selecting different features for different samples.

Finally, Figure 12 displays examples of CIFAR-10 predictions given different numbers of revealed patches. The predictions generally become accurate after revealing only a small number of patches, reflecting a similar result as Figure 3. Qualitatively, we can see that the policy network learns to select vertical stripes, but the order in which it fills out each stripe depends on where it guesses important information may be located.

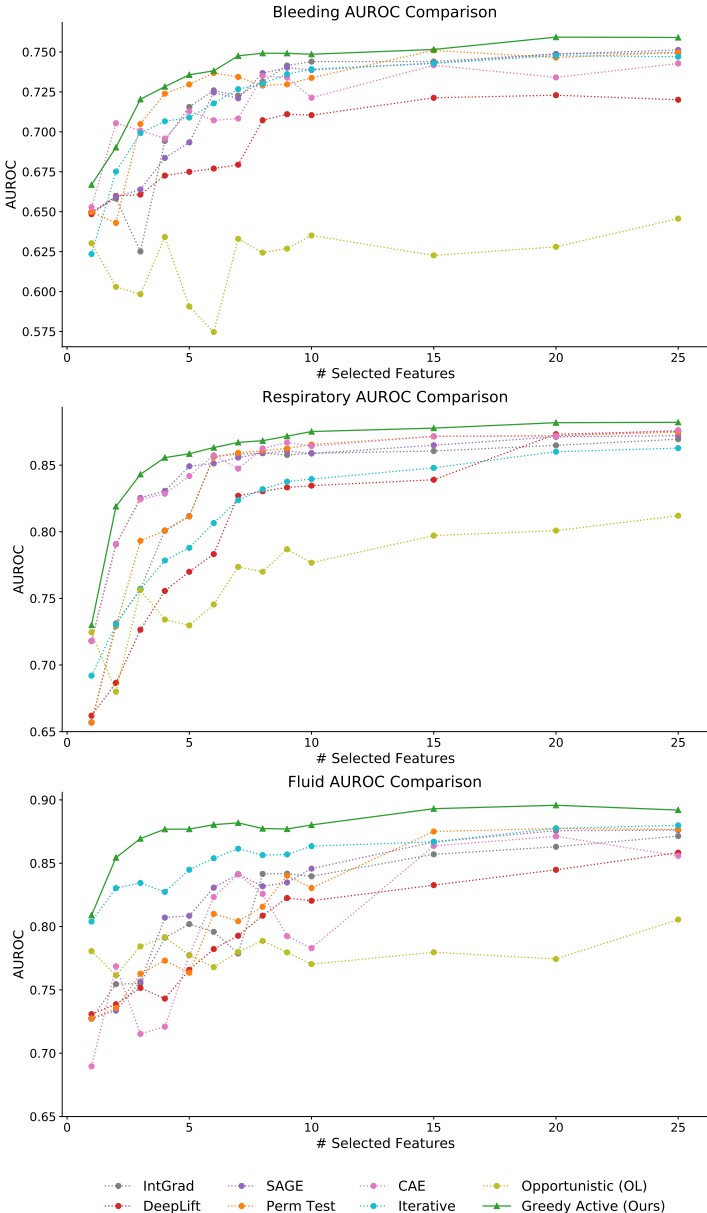

Figure 4: AUROC comparison on the three emergency medicine diagnosis tasks.

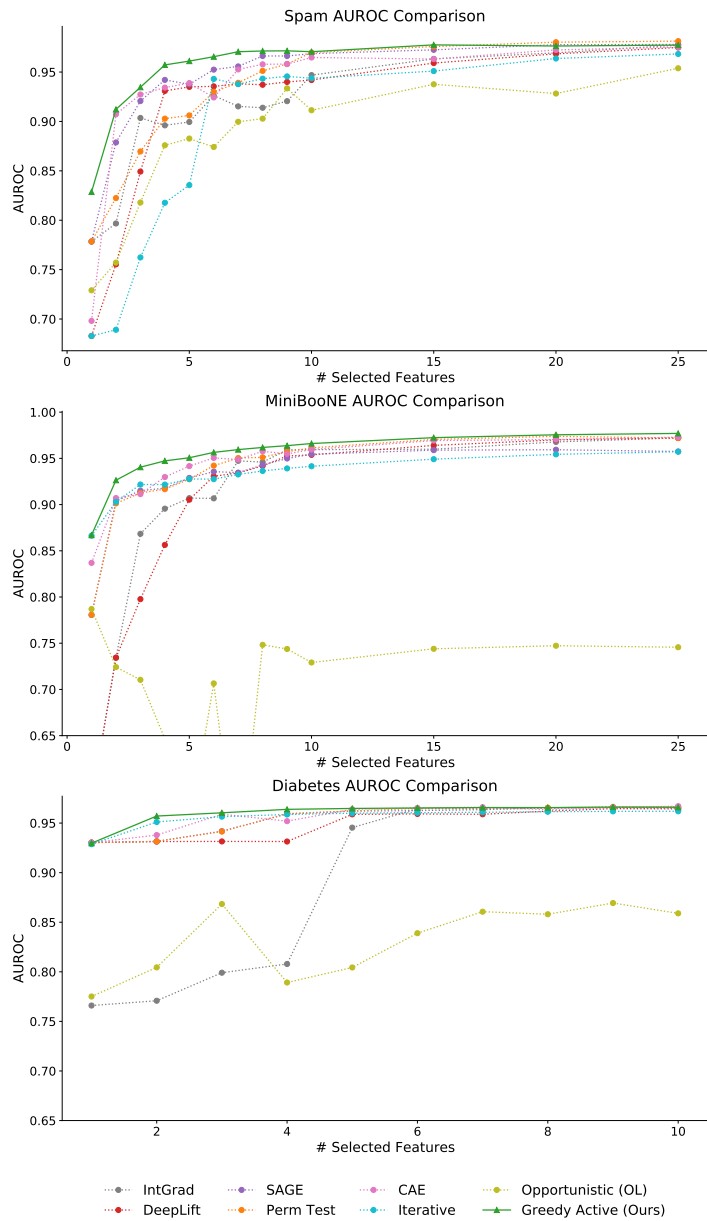

Figure 5: AUROC comparison on the three public tabular datasets.

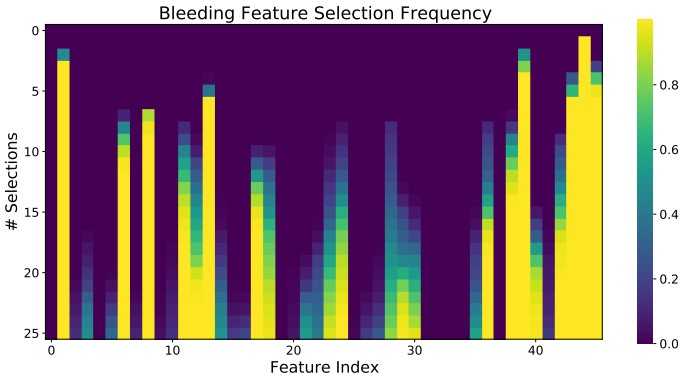

Figure 6: Feature selection frequency for the greedy active approach on the clotting dataset.

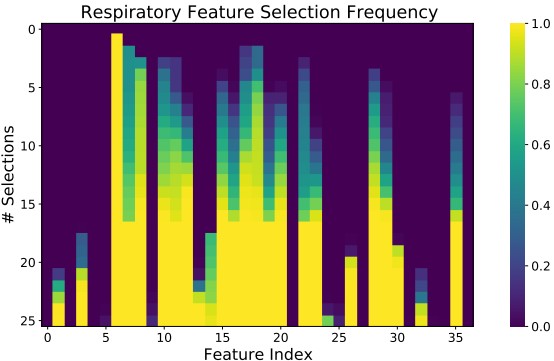

Figure 7: Feature selection frequency for the greedy active approach on the respiratory dataset.

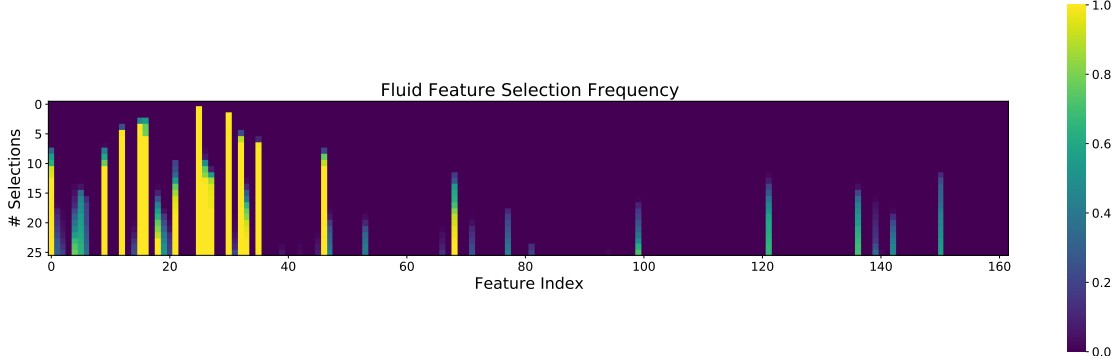

Figure 8: Feature selection frequency for the greedy active approach on the fluid dataset.

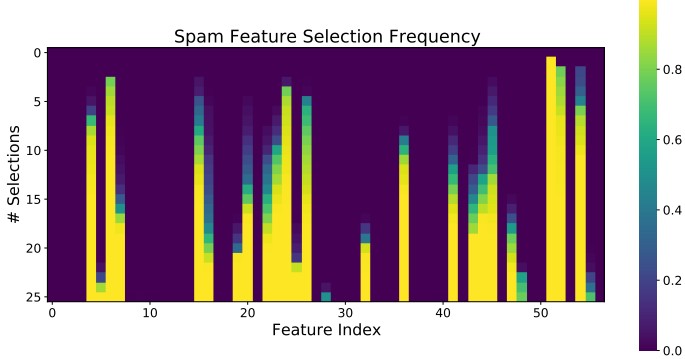

Figure 9: Feature selection frequency for the greedy active approach on the spam dataset.

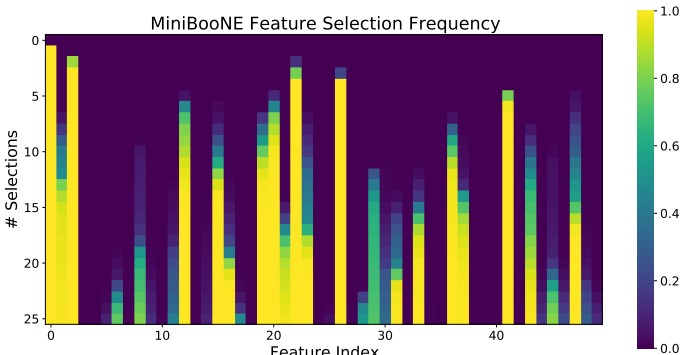

Figure 10: Feature selection frequency for the greedy active approacg on the MiniBooNE dataset.

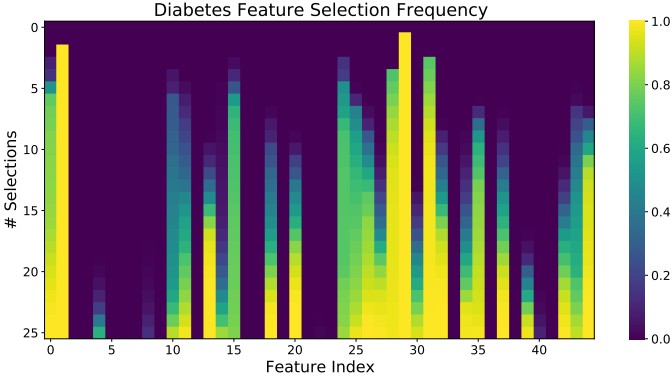

Figure 11: Feature selection frequency for the greedy active approach on the diabetes dataset.

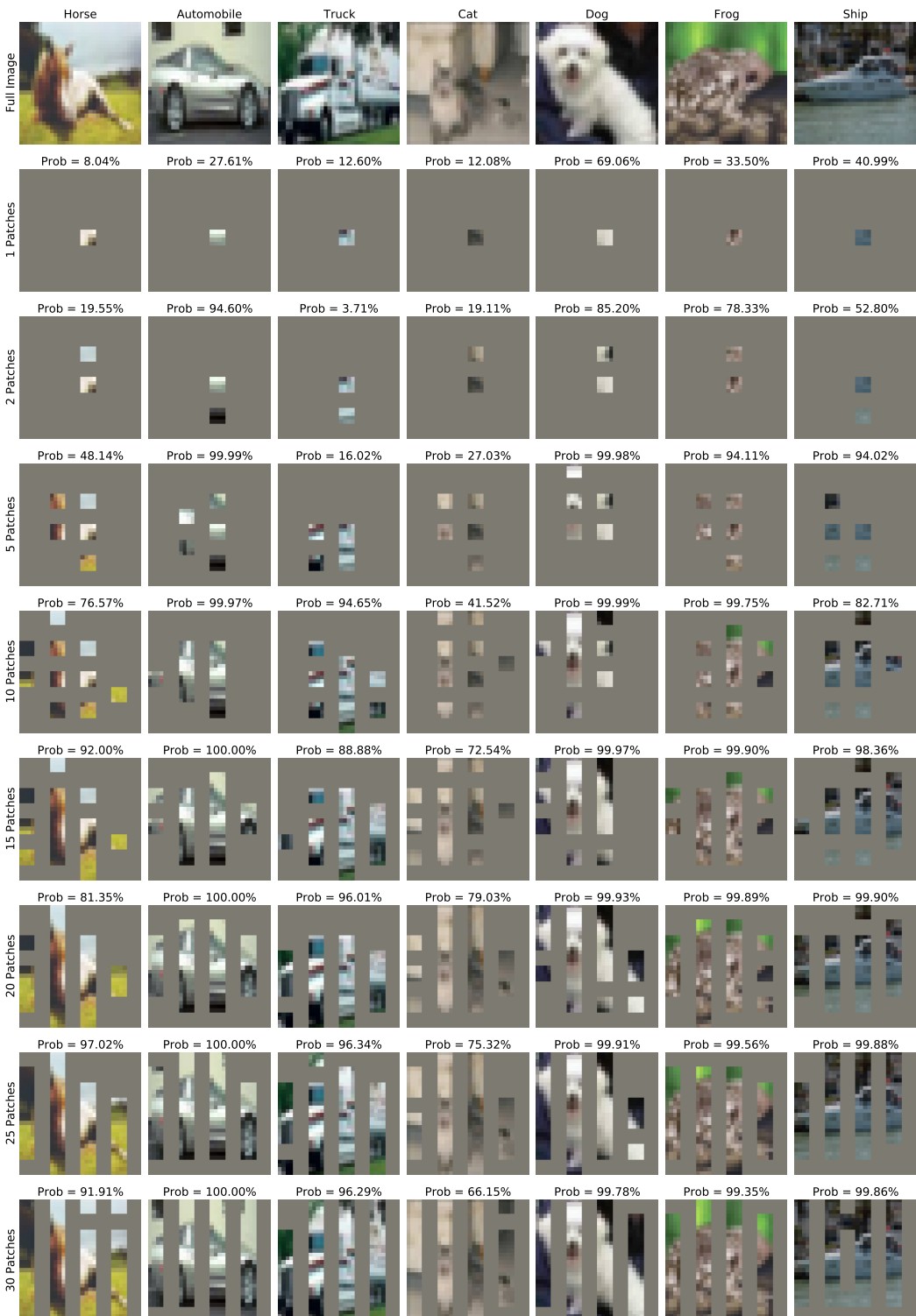

Figure 12: CIFAR-10 predictions with different numbers of patches revealed by our greedy active method.

