# OpenReview forum: "Greedy Information Maximization for Online Feature Selection"
_ICLR.cc/2023/Conference — Submitted to ICLR 2023_

### Official Review · Reviewer_dhvL · 2022-10-24

**Confidence:** 3
**Correctness:** 4
**Technical Novelty And Significance:** 3
**Empirical Novelty And Significance:** 3
**Recommendation:** 6

**Clarity, Quality, Novelty And Reproducibility:**

In general, the paper has a complete structure, clear thinking and some innovation.



**Strength And Weaknesses:**

Strength:
1. Starting from the greedy algorithm, the author continuously analyzed the algorithm from the theoretical and strategic levels, so as to get their proposed algorithm, which I think is very reasonable.
2. The effectiveness of the proposed method was demonstrated on various datasets. The authors also provide their codes of experiments.
Weaknesses:
1. I think the explanation of eq.(8) is far from sufficient. It's better to restate how far it differs from eq.(1) and how do you derive this formula from the greedy algorithm in more detail.
2. In section 3, authors say that 'The third example illustrates the perils of following a greedy approach: it fails to account for each
selection’s impact later in the selection procedure. In contrast, a non-greedy policy can make suboptimal selections that yield large improvements in later steps.' Does that show up in the experimental section? If not, can you design it?
3. In Figure 3, a visualization of the selected features is given, which I think is very good. But I think it would be better if there was a reasonable explanation for the chosen features.








**Summary Of The Paper:**

This paper focus on the online feature selection problem which mainly viewed as a reinforcement learning problem by traditional methods. Authors introduced a greedy approach to online feature selection that selects features according to their conditional mutual information with the response variable and then proposed a deep learning approach to learn the greedy policy.  Experiments on numerous datasets show that the proposed method outperforms a variety of existing feature selection methods.

**Summary Of The Review:**

Overall, the article has a complete structure and clear thinking. But the content needs to be further improved, if authors can solve my problems, I think this paper is a very good article.

---

> ### Author Response · Authors · 2022-11-19
> **Reviewer dhvl response**
>
> We would like to thank the reviewer for closely examining our work and providing their feedback. We have responded to many questions/concerns in our general response, and we have used to space below to respond to your remaining questions:
>
> **Explanation for eq. 8.** In your review, you asked for more clarity around how eq. 8 is derived from eq. 1. There is a key difference between these, which is that eq. 8 does not optimize over the entire trajectory, but instead treats the problems independently for each subset. This is what makes our approach greedy rather than an RL-based solution. We have clarified this by adding the following sentence to Section 4.1: “Our objective thus optimizes for individual selections and predictions rather than the entire trajectory, which lets us build on earlier results from Section 3.”
>
> **Scenario in Example 3.** In your review, you asked if the scenario illustrated in Example 3 (the xor problem) shows up in the experiments. Our experiments focused exclusively on real datasets, and we cannot know for certain if such a relationship exists in the data. It would be possible to create a synthetic dataset with this relationship, but again, we are interested primarily in realistic datasets. Beyond the scope of our work, it is often the case that greedy algorithms fail in contrived scenarios but work reliably in practice, and that’s why we think it’s useful to develop a greedy approach for this problem.
>
> **Understanding selections in image experiment.** Thanks for this suggestion, we’ve added an attempt at an explanation for the image patch selections in Appendix E.

---

### Official Review · Reviewer_H6xQ · 2022-10-25

**Confidence:** 4
**Correctness:** 3
**Technical Novelty And Significance:** 2
**Empirical Novelty And Significance:** 2
**Recommendation:** 5

**Clarity, Quality, Novelty And Reproducibility:**

After initially reading the title and abstract of this paper, I thought this paper presented an interesting method for online feature selection. However, after reading the main text, I felt a little disappointed.

I have the following comments/questions. I look forward to the response/clarification from the author(s). Thanks.

---

### Clarity:

1. Some of the descriptions in this paper are not so clear. Especially the description of mathematical notations, many places in the paper are not clearly explained at the beginning. To understand the meaning of mathematical notions, one must read the following content first. I don't think this is the way a normal/good paper should be presented. For example, on Page 2, Section 2.1, Line 2 in the 1st Paragraph, maybe it needs to explain "$\mathcal{X}_i$"; on Page 2, Section 2.1, Line 2 in 2nd Paragraph, maybe it needs to explain "$\pi(\cdot)$"; The last sentence in 1st Paragraph of Section 2.2, for $k$, it is better to explain the relationship between $k$ and $d$; and so on.

2. For the statement below Eq. (1), "...goal in designing a policy is to minimize... or to maximize our final predictive accuracy". Are these two descriptions equivalent in terms of classification?

3. On Page 1, the 1st Paragraph of Section 1, the last sentence, "...We refer to this problem as online feature selection, and it has been considered by several works in the last decade...". A quick Google search can find that the review of the current online feature selection methods is insufficient. There are many new developments in online feature selection (including 2021 and 2022). Some works are even more advanced than the work in this paper, such as dealing with online feature selection for multiple sources. In addition, the review of the offline feature selection methods is also insufficient.

4. When comparing offline feature selection methods, why use "a supervised version of CAE" (which is itself unsupervised learning) for comparing offline feature selection methods instead of the STG method (which is itself supervised learning)?

5. I do not quite agree with the processing of calculation results in this paper. For example, in Table 1, "the mean AUROC across k = 1... 10 features", in fact, for feature selection based on deep neural networks, the stability of selected features is one of the major potential issues (In Figure 5 of the STG paper [1], the authors specially analyzed the stability of selected features). So, the author(s) should present (multiple) cross-validation results with a fixed $k$ value (mean +/-standard error). Also, the curves in Figures 2 and 3 should be the result of multiple cross-validations with a fixed $k$ value (mean +/-stance error). In this way, it can reflect the stability of the selected features.

---

### Novelty

I think the novelty of this proposed algorithm in this paper is quite limited.

(1) Although the topic of this paper is online feature selection, the main idea of this paper, in my opinion, it only replaces the feature selection layer in STG [1] with the feature selection layer in concrete auto-encoders (CAEs) [2].

(2) When CAEs appeared, the concrete distribution was combined with feature selection for the first time, which was novel and eye-catching. But since then, several studies have used this method for feature selection. So, now it is better to simplify Section 4.2 "CONTINUOUS RELAXATION" and move the superfluous/cumbersome content to the Supplementary Materials.

In addition, some other tiny issues/typos

(1) In general, the first time an abbreviation appears, it needs to go with the full name. So, please give the full name of "RL" in the Abstract.

(2) The format of the references is quite inconsistent. Please check carefully and correct it.

---

[1] Yamada Y, Lindenbaum O, Negahban S, Kluger Y. Feature selection using stochastic gates. In International Conference on Machine Learning 2020 Nov 21 (pp. 10648-10659). PMLR.

[2] Balın MF, Abid A, Zou J. Concrete autoencoders: Differentiable feature selection and reconstruction. In International conference on machine learning 2019 May 24 (pp. 444-453). PMLR.

---





**Strength And Weaknesses:**

### Strength:

Feature selection is a significant branch of machine learning, and most of the existing feature selection studies mainly focus on offline feature selection. In practice, online feature selection is more challenging and more important.

---

### Weaknesses:

1. The novelty of this proposed algorithm is limited.

2. The review and comparison with online/offline feature selection are insufficient.

3. There are potential issues in the calculation of experimental results (especially the lack of stability analysis for selected features, which is necessary for deep model-based feature selection).

4. Some descriptions in this paper are not clear.

For more details, please see the section of "Clarity, Quality, Novelty And Reproducibility"

---




**Summary Of The Paper:**

In this paper, the author(s) considered the problem of online feature selection which was different from the standard feature selection with fixed feature subsets (i.e., offline feature selection). The author(s) proposed a deep learning-based algorithm to process the greedy information maximization for online feature selection. Then, the author(s) demonstrated that empirical experiments showed the advantage of the proposed algorithm.

---


**Summary Of The Review:**

The work in this paper is interesting; However, there are many unclear aspects in the description (including experiments) of this paper, and the innovation also needs to be further discussed. In addition, the work largely ignores existing works.

---

> ### Author Response · Authors · 2022-11-19
> **Reviewer H6xQ response**
>
> We would like to thank the reviewer for closely examining our work and providing their feedback. We have responded to many reviewer questions/concerns in our general response, and we have used to space below to respond to your remaining questions:
>
> **Overview of offline/static feature selection.** In your review, you mentioned that you thought our overview and comparisons were insufficient. Regarding the overview, static feature selection really isn’t the subject of our work so we can’t provide a complete overview, but we have added references to multiple relatively recent review papers and a couple more recent differentiable feature selection papers. Regarding comparisons, we wanted to focus on state-of-the-art methods that work with neural networks, and to our knowledge the CAE (and perhaps STG, we address this below) is SOTA. From reviewing the literature and looking at the baselines in these works, it doesn’t seem like we’re missing any critical methods.
>
> **Review of online feature selection methods.** We have mentioned several additional works throughout our paper, our review should now be sufficient. Our overview is based on having checked the citations of several of the most recent works on this topic. We would also like to point out that any reviews you found for “online” feature selection are likely related to a different problem; while the term is used in some prior works, we’ve found that the preferred name for our problem is “active feature acquisition” and we’ve adjusted the paper accordingly.
>
> **CAE vs. STG.** In your review, you asked why we used the CAE instead of the STG as an offline/static baseline. You seem very familiar with both methods, but the key points for our decision were:
>
> - The STG method penalizes the number of selected features, whereas the CAE sets a specific constraint. This means that to use STG in our experiments, we would need to identify lamba values that yield each specific number of features, which would be quite laborious given the number of datasets and budgets we tested. The difficulty of choosing a lambda value is further compounded by the stochasticity of training: the same lambda value can yield different numbers of features on different runs.
> - The fact that the CAE is unsupervised by default isn’t too important – it’s easy to change the prediction target and loss function, thus making it supervised. The CAE paper even pointed this out, so what we’ve done here isn’t very exotic. STG is supervised by default so we would have been happy to use it, but given the reasons above we found it easier to use the CAE.
>
> To alleviate any possible concerns on this topic, we conducted experiments with STG using the authors’ code, and we compared it with the supervised CAE on MNIST. We found 1) that it was difficult to identify lambda values that yield specific numbers of features, 2) that the number of features could change due to the random seed even with a fixed lambda value, and 3) that even when we did match the desired number of features, the performance was nearly identical to the CAE. Given all of this, we decided to stick with the CAE.
>
> **Novelty vs. STG/CAE.** In your review, you mentioned that you thought our approach was basically replacing the STG selection layer with the CAE selection layer. We addressed this point in our general response, but briefly the differences are 1) our layer selects just one feature at a time vs. multiple features in CAE/STG, 2) our layer conditions on subsets of available features vs. unconditional selections in both CAE/STG, and 3) our two networks create a more challenging training problem than both CAE/STG. The differences thus go far beyond the fact that both our work and the CAE use the Concrete distribution.

---

### Official Review · Reviewer_84q8 · 2022-10-26

**Confidence:** 4
**Correctness:** 3
**Technical Novelty And Significance:** 2
**Empirical Novelty And Significance:** 2
**Recommendation:** 3

**Clarity, Quality, Novelty And Reproducibility:**

The paper is written somewhat clearly -- the algorithm used in the experiments
is somewhat hard to parse given the semi-related theorems in Section 3 and the
relaxation in Section 4.2 that alters the "ideal" greedy step in Equation (8).
The main ideas could be delivered more directly.
The novelty of this work seems limited since conditional mutual information is
known to work well for feature selection. The main contribution of this work is
using RL and differentiable masking to find the unselected feature in a greedy
manner. The experiments appear to be reproducible (I checked the supplementary
material).

**Strength And Weaknesses:**

**Strengths:**
- Example 3 is a nice instance that shows the difficulty of feature
  selection, especially when using pairwise feature information.
- The experiments look *very strong*. Other methods seem somewhat noisy and
  "less monotonic". How many trials were used for the different methods -- both
  feature selection and model trainings?

**Weaknesses:**
- The paper bounces back and forth between "combinatorial" and RL-based feature
  selection.  It would be beneficial to present the main algorithm as directly
  as possible (possibly by just explaining all the details of Figure 1): how
  exactly do you use RL, what is the differentiable masking expression, is the
  criteria ultimately that of the downstream model, etc. There are many more
  details and asides that take away from the empirical results.
- Is Theorem 1 actually true? In the hidden XOR case if $y = x_1 \otimes x_2
  \otimes x_3$ and $d = 10$, then in the second step of feature selection, all
  candidates are equally good (i.e., we don't need to put all the mass on one
  feature). Maybe say "a global optimum" in the event of ties.
- This work isn't really about "online feature selection" -- it's just adaptive
  and "normal" supervised feature selection.
- In Section 6.1, you have "In each of these problems, gathering all possible
  inputs is impractical due to time and resource constraints, thus making
  online feature selection a natural solution." How do you learn the RL policy
  then? It seems that you need all of the features since you learn a mask on the
  unselected features via RL and the downstream model.
- Table 1: Why is the mean loss for different feature set sizes a meaningful
  quantity? It carries some signal, but these values are better understood when
  plotted as a Pareto curve as in Figure 2.
- It would be valuable to compare your RL-based greedy algorithm to the true
  greedy algorithm that compute the conditional mutual information in each step
  (relative to set $s$). Of course, this can be too expensive to compute in
  practice, but it would be valuable to know ``what is achievable'' as an upper
  bound and how close your algorithm comes to that.

**Suggestions:**
- [page 2] Suggestion: The conventional notation is to use $S \subseteq [d]$ for
  a subset of features (i.e., capital $S$).
- [page 4] It would be worthwhile to discuss how $I_i^n$ in Equation (5) is
  related to the empirical conditional mutual information on the training data.
- [page 7] Suggestion: The related works section would go better at the end of
  the introduction.
- [page 7] In the related works when discussing "iterative feature scoring
  methods", it would be good to cite the landmark paper "Fast binary feature
  selection with conditional mutual information" (Fleuret JMLLR, 2004) for the
  CMIM algorithm, since this is a strictly combinatorial version of the (binary)
  cross entropy algorithm you propose.
- [page 7] How do you differentiate online feature selection vs "offline
  feature selection". As far as I can tell, these problems are equivalent and
  best described as "feature selection" or "supervised feature selection".
- [page 8] Suggestion: Include the standard deviation of each value in Table 1.

**Summary Of The Paper:**

This work studies an RL-based greedy feature selection algorithm that uses
conditional mutual information with the label as the scoring criteria. The
authors propose using a differentiable method in each greedy step by combining
the true model and the "concrete distribution" (for feature masking) to learn
the best policy given the current set of features, i.e., to identify the next
best feature. They provide some theoretical justification for their approach,
and give a diverse set of feature selection experiments that suggest their
method is both effective and practical.

**Summary Of The Review:**

This paper is on to something, especially if the experiments are dominant
across many averaged trials and model trainings. That said, I think the paper's
message could be greatly simplified by very directly presenting the algorithm
and not justifying all of the decisions with lemmas about mutual information,
since most these ideas are reasonably well-known. I recommend the paper be
rejected from ICLR 2023, but encourage the authors to refine the work and
resubmit to a comparable venue in the future.

---

> ### Author Response · Authors · 2022-11-19
> **Reviewer 84q8 response**
>
> We would like to thank the reviewer for closely examining our work and providing their feedback. We have responded to many questions/concerns in our general response, and we have used to space below to respond to your remaining questions:
>
> **Not RL.** It seems from your review that you thought our approach was somehow using RL. We’d like to emphasize that our approach is actually an alternative to RL, which should be easier to learn because it’s simply a sequence of locally optimal decisions. The paragraph at the end of Section 4.1 addresses this point. For example: “RL methods generally face a challenging exploration-exploitation trade-off, but learning the greedy policy is simpler because it only requires finding the locally optimal choice among $\mathcal{O}(d)$ options at each step.”
>
> **Problem formulation.** Your review asked us to further clarify in what sense our problem is online/adaptive rather than standard supervised feature selection. In standard supervised feature selection (what we previously referred to as “offline” but now refer to as “static”), the goal is to identify a single feature set that is small but retains high predictive power. In our setting, the feature set can differ between examples because features are selected sequentially. Multiple reviewers were confused by the term “online,” so we have replaced the problem name with “active feature acquisition” throughout the paper.
>
> **Fleuret (2004).** Thank you for pointing us to this work, which we had mistakenly overlooked in our original submission. Like you said, this paper also aims to maximize CMI, but it makes important simplifying assumptions: the assumptions are 1) that the features are binary (see their section 2.1), and 2) that the CMI can only be calculated when conditioning on a single feature (see their eq. 2). These limitations were natural for the time, and even today accurately estimating CMI is difficult. Our approach sidesteps the estimation problem by amortizing the selection logic into a policy network, which recovers the CMI maximization approach when it’s perfectly optimized. Because our selections are made by a learned network, our approach has the benefits that we can a) condition on multiple features, b) work with continuous-valued features, and c) make selections in a single forward pass (vs a slow iterative scoring procedure). We have clarified these differences in the paper (see Section 3.2 and Section 5).
>
> **Other CMI methods.** Your review requested that we shrink Section 3 to de-emphasize the CMI derivation, which is to some extent known in the literature. These results need to be included because our learning approach ultimately builds on them: they are useful for formulating our amortized optimization approach that recovers the greedy policy when perfectly optimized. We would also like to re-emphasize the key difference in our work compared to others that discuss CMI – that we make this approach simple to implement and effective in practice.
>
> **Empirical conditional mutual information.** In your review, you asked us to compare $I^n_i$ to the empirical CMI. We are not sure if you’re referring to a specific estimator that has this name, but we clarified that our procedure should be understood as approximating the CMI: “Before introducing our main approach (Section 4), we first describe a procedure to estimate the conditional mutual information given access to the response distributions $p(y \mid x_s)$ for all $s \subseteq [d]$ and the conditional distributions $p(x_i \mid x_s)$ for all $s \subseteq [d]$ and $i \in [d]$.” It’s basically a CMI estimator that relies on oracle access to the joint data distribution.
>
> **Comparison with true CMI.** Your review also suggested comparing our approach to selections based on the exact CMI. As you might expect, estimating the CMI is challenging for any real dataset, and as clarified above our iterative procedure can be understood as a CMI estimator. Thus, the comparison is in some sense already provided. One further option would be to add a synthetic dataset where the CMI is known a priori, but we think this is not high-priority compared to experiments with real datasets.
>
> **Emergency medicine datasets.** Your review pointed out that if obtaining all features is too costly, we won’t have the data necessary to train our method. This is correct – our method (like most others for this problem, as well as those for standard/static feature selection) requires all features during training. Eliminating the need for this training data would also be interesting, but our focus is only reducing feature acquisition cost at inference time. All the datasets used in our experiments are therefore fully observed, i.e., the training data contains values for all features.

---

### Official Review · Reviewer_h13P · 2022-10-29

**Confidence:** 4
**Correctness:** 4
**Technical Novelty And Significance:** 1
**Empirical Novelty And Significance:** 3
**Recommendation:** 3

**Clarity, Quality, Novelty And Reproducibility:**

The paper is clearly written.

Greedy strategy is very often used in practice. So the suggestion to use it to maximize the info gain is not really novel. Plus i have concerns in the foundational aspects that the authors use to motivate their method as listed in the weakness section.

No concerns about reproducibility

The empirical results are promising, and add to the quality of the paper. However I would argue overall the quality is below this conference's standards.

**Strength And Weaknesses:**

Strengths:
--- Clear and easy to read
--- Empirical evaluations are interesting and useful.


Weaknesses:
--- The paper lacks solid theoretical justification. They cite Das/Kempe 2011 who analyze linear regression functions, but miss citing some other relevant literature that very likely covers their cost functions.

General cost functions:
Restricted strong convexity implies weak submodularity. Elenberg et al. Annals of Stats

Streaming feature selection:
Streaming Weak Submodularity: Interpreting Neural Networks on the Fly. Elenberg et al. Neurips 2017
Online Streaming Feature Selection. Wu et al. ICML 2010

--- While the authors have tried to cite many relevant works, the citations on feature selection are still limited. For example, gradient
based feature selection is also covered in Elenberg et al works above and other works such as:
Fast Feature Selection with Fairness Constraints. Quinzan et al.

--- I am not sure why modeling p(x_i | x_s) as explained in Sec 3.2 is even important. Sec 4 onwards the authors explain ways to circumvent it. But many other methods (some of which are listed above) only care about the correlation with y when selecting features (along with iterative strategies that ensure the selected features are uncorrelated amongst themselves). This also obviates the development in Sec 4.



--- Other than the above point, I am not sure why E_s in the cost function is a good idea, especially in a greedy strategy. Why would one want to take an expectation over ALL subsets of size s when we care only about a particular set as formed by the greedy strategy? Is that not a gross overkill ? It seems it is done for a theoretical justification, but again, it seems unfounded and not really required given theoretical justification based on weak submodularity.

--- The section on greedy suboptimality is interesting, but I think it is also superseded by the weak submodularity papers.




**Summary Of The Paper:**

The paper proposes a greedy strategy to feature selection based on information maximization criterion.

**Summary Of The Review:**

I think there are concerns about theoretical justifications and the authors missing relevant works when motivating their method. The empirical results are interesting, but not surprising given the generally strong performance of the greedy method. The paper is not ready to be published at a conference like this one.

---

> ### Author Response · Authors · 2022-11-19
> **Reviewer h13p response**
>
> We would like to thank the reviewer for closely examining our work and providing their feedback. We have responded to many reviewer questions/concerns in our general response, and we have used to space below to respond to your remaining questions:
>
> **Relevance of (weak) submodularity.** Thanks for your comments and references related to weak submodularity and its applications to feature selection. We were aware of the relevance given our citation of Das & Kempe (2011), but we were not familiar with all of the works you mentioned – we’ve added several new citations to Section 3.3. However, it’s important to point out that (weak) submodularity doesn’t apply in this context and can’t prove anything about the suboptimality of the greedy approach; we cited Das & Kempe (2011) mainly as an analogy for what we would want to prove here. The reason (weak) submodularity doesn’t apply is that our problem doesn’t have a fixed set function (the set function changes with each selection) and feasible solutions aren’t represented by subsets (they’re represented by policies). This is known in the literature, and we’ve updated our paper to mention that the notion of “adaptive submodularity” which could *potentially* apply here (see Golovin & Krause, 2011) actually doesn’t. We also came across one paper that manages to derive suboptimality bounds under strict distributional assumptions (Chen et al., 2015). Our overview of related work has been greatly improved by looking into this more, so thank you for your comments.
>
> **Why modeling $p(x_i \mid x_s)$ matters.** In your review, you asked why it’s important to model the feature conditional distributions and why we wouldn’t care only about relationships between individual features $x_i$ and the response $y$ (e.g., correlation). This is simple: if you don’t model feature relationships, you may end up selecting features with redundant information. The most obvious case is where $x_1$ and $x_2$ are highly correlated and both provide information about $y$: after selecting just one of them, you may be better off selecting a different feature even if it has lower $H(y \mid x_i)$. This redundancy is captured by CMI, which implicitly requires modeling $p(x_i \mid x_s)$. We've added a citation for a method that does this using a modified VAE (Ma et al., 2018), and emphasized that our approach for circumventing this is valuable because of the difficulties in training effective generative models.
>
> **Novelty of greedy strategy.** In your review, you wrote that it’s not surprising to see our method work well given the generally strong performance of greedy methods. Greedy algorithms work well in general across many problems, but as far as we can tell, there isn’t currently a reliable implementation of the greedy CMI approach for this problem. With that in mind, it seems like a reasonable contribution to create a greedy method that’s simple to implement and works in this context.

---

### Official Review · Reviewer_oRCJ · 2022-10-31

**Confidence:** 3
**Correctness:** 2
**Technical Novelty And Significance:** 3
**Empirical Novelty And Significance:** 3
**Recommendation:** 5

**Clarity, Quality, Novelty And Reproducibility:**

The majority of the paper is well-written, and I like the idea of the proposed method. However, I didn’t clearly see where this method is “adaptive” from its theory and experiments, alleged in the abstract. I still need more details to understand part of the equations and proof (which could be due to my misunderstanding). The notations can be improved especially in distinguishing between sets and scalars.

**Strength And Weaknesses:**

Strengths:
- The method is novel based on its description. The solution is interesting.
- The writing is of high quality
- Experiments well demonstrated the proposed method.

Weaknesses:
- One weakness is the clarity of the notations. Although the author defined the notations at the beginning of section 2. I found I still got lost while reading the paper, especially when trying to distinguish between an index set and a single index. This could be due to the redefinition of the bold type and the normal type.
- About the clarity of the problem definition: I’m not familiar with this task, but when people refer to “online,” they usually will indicate the data distribution changes or is stationary. I guess here the problem assumes a stationary distribution, and “online” indicates the sequential nature of the selection, right? If so, there is no adaptive part of the problem, and the method sounds like a sequential version of the offline counterpart.
- For the theoretical results, I feel Eq. 8 and thm 1,2 may require more explanations or assumptions. (But these impressions could also be due to my misunderstanding.) I had the following questions while I was reading the paper
  1. Does p(s) depend on $\phi$ or $\theta$?
  2. In the proof of Thm 1, what are the mild conditions mentioned in the main paper? I didn’t follow how Eq. 8 simplifies to Proposition 1.
- It seems the practical objective function of the experiments has a gap with Eq. 8. I.e., I didn’t see how to sample from $p(s)$ from Alg. 1. It would be great to state the objective function in experiments (with monte carlo estimates of the expectations).
- Should Eq. 7 have a reverse inequality sign because $v$ is the expected loss? This is the same for the gap definition below.
- I felt some parts of section 3 (mainly sec. 3.2) are pretty technical but didn’t help me understand the main method in section 4. Perhaps leaving more space in section 4 to explain the training procedure is more rewarding to the audience. (Maybe it’s only a personal taste.)


**Summary Of The Paper:**

The paper introduces a new greedy method for the online feature selection problem. The method is locally optimal in selecting features that maximize the conditional mutual information with the response variable. The authors propose a deep learning-based approach to learning a selection policy network, which is trained by amortized optimization. The new method is of low complexity compared to other RL-based methods and has good performance compared to online and offline methods on tabular and image data.

**Summary Of The Review:**

This work presents a smart way to make online feature selection, which is novel given the literature. The paper is also well-written, and the experiments demonstrated the method well. But as I mentioned above, I’m still curious about the adaptive part of the problem definition. The main results may require more explanations or assumptions.

---

> ### Author Response · Authors · 2022-11-19
> **Reviewer oRCJ response**
>
> We would like to thank the reviewer for closely examining our work and providing their feedback. We have responded to many questions/concerns in our general response, and we have used to space below to respond to your remaining questions:
>
> **Problem definition.** Your review mentioned that you were not sure about our problem formulation. As you stated, our problem is basically a sequential version of supervised feature selection (with no special consideration for shifting data distributions). It seems the term “online” was confusing for multiple reviewers, so we have replaced the problem name with “active feature acquisition” throughout the paper. To be clear, this means separately identifying predictive features for each data example.
>
> **Questions about eq. 8.** Your review raised several important points about our objective and training procedure, and we’ve made sure to address these in the main response. We would like to emphasize that nothing about our approach has changed (because the original formulation has the correct global optimizer), but our presentation has improved in several places. Perhaps most importantly, when addressing the choice to use the policy rather than $p(s)$ to generate feature sets, we have edited the text as follows: “Rather than training with a random subset distribution $p(s)$, we generate subsets using features selected by the policy $\pi(x; \phi)$. This allows the models to focus on subsets likely to be encountered at inference time, and it does not affect the globally optimal policy/predictor: gradients are not propagated between selections, so both eq. (8) and this sampling approach treat each feature set as an independent optimization problem, only with different weights (see Appendix D).”
>
> **More space for training procedure.** In your review, you asked if we could allocate more space to Section 4 and less to Section 3. Section 3 has become slightly shorter, and we have reframed it as providing preliminary results (Propositions 1 and 2) that are necessary to develop our amortized optimization approach. Section 4 has also become slightly longer due to some additions, and hopefully also clearer due to us responding to several reviewer questions.
>
> **Reverse inequality in eq. 7.** In your review, you asked if we should reverse the inequality sign used in eq. 7. The inequality is correct: the optimal policy-predictor pair will achieve a lower loss than the greedy approach, so that $\min_{\pi, f} v_k(\pi, f) \leq v_k(\pi^*, f^*)$.

---

### Official Review · Reviewer_d6k9 · 2022-11-02

**Confidence:** 3
**Correctness:** 3
**Technical Novelty And Significance:** 2
**Empirical Novelty And Significance:** 3
**Recommendation:** 5

**Clarity, Quality, Novelty And Reproducibility:**

The organization is good and the logic is clear. There should not be many issues with reproducibility.

**Strength And Weaknesses:**

Strengths:
+ The problem is interesting and meaningful. Feature selection, especially online feature selection, could be used in various application scenarios in practice.
+ The empirical evaluation is comprehensive and overall convincing. The paper conduct extensive experiments on several datasets including tabular datasets and image classification datasets, and the proposed method outperforms both RL-based and offline feature selection methods. The authors will release the source code as mentioned in Section 6.

Weaknesses:
- The novelty is limited. Though the framework is novel, the individual models are not.
- The diagram in Figure 1 is interesting. However, the key point of the proposed method is not reflected in Figure 1.  More information could be added to the diagram.
- Some typos should be modified. For example, in Subsection 2.1, “X_{\hat{s}} is the set complement X_{[d]\s}” -> “X_{\hat{s}} is the set complement to X_{s}”.
- Some details could be improved. For example, the “eq.”  could be changed to “Equation”  in order to be consistent with the description “Figure”, “Table”, etc.

**Summary Of The Paper:**

This paper proposes a greedy-based approach for online feature selection.  This paper defines greedy online feature selection and also provides an iterative procedure to implement the greedy approach with a deep learning approach.


**Summary Of The Review:**

The paper proposes a new greedy-based method to address online feature selection. The technical novelty in the individual parts is limited.

---

> ### Author Response · Authors · 2022-11-19
> **Reviewer d6k9 response**
>
> We would like to thank the reviewer for closely examining our work and providing their feedback. We have responded to many questions/concerns in our general response, and we have used to space below to respond to your remaining questions:
>
> **Novelty of individual models.** In your review, you wrote: “The novelty is limited. Though the framework is novel, the individual models are not.” The problem we’re addressing ultimately comes down to selecting features and making predictions, so the individual models cannot be much more sophisticated than what we have here (a policy and predictor network). The more interesting part of this problem is determining how to train the models, and since you agree that our framework is novel, it would seem that our training approach represents a worthwhile contribution.
>
> **Figure 1.** In your review, you wrote that Figure 1 does not reflect key parts of the training procedure. It’s intended to be an accessible summary of the training approach, and any remaining details are reflected in our training pseudocode (“Our training procedure is summarized in Figure 1 and Algorithm 1”). We tried moving Algorithm 1 to the main text, but unfortunately this wasn’t possible given the space constraints.
>
> **Typos.** Thanks for pointing out that typo – we’ve closely reviewed the paper to ensure that there aren’t any others.
>
> **Correctness of claims.** We noticed that you rated our claims as having minor issues, but you didn’t mention any such issues in your review. We’ve addressed several specific points raised by other reviewers, but we would be happy to discuss any remaining questions/concerns you have about our theoretical results.

---

### Author Response · Authors · 2022-11-19
**General response (1/3)**

We would like to thank all six reviewers for closely reading our work and providing their feedback. There were numerous helpful suggestions provided by all reviewers and also some confusion regarding our problem formulation, and we have attempted to incorporate all of this feedback in our revisions. Overall, we’ve made numerous writing improvements that we hope will greatly improve your opinion of the paper. On the other hand, the method itself has remained unchanged because all the concerns about it seemed related to our presentation. We hope that the reviewers can read our response and update their reviews as appropriate.

First, to briefly summarize the paper, our goal is the following. In settings where features are selected one at a time (or sequentially), we aim to determine 1) which features to select next and 2) how to make predictions given the currently available features. We propose addressing this with a greedy approach, where features are selected according to their conditional mutual information $I(x_i, y; x_s)$. This approach is theoretically appealing and has been discussed by some prior works, but it’s difficult to implement in practice. We propose a procedure to learn policy and predictor networks that approximate the greedy approach, in an end-to-end fashion and without modeling feature conditional distributions $p(x_i \mid x_s)$. This greedy approach might be expected to underperform RL-based methods that optimize across the entire selection process, but we observe that it can actually work better in practice.

There are many points to address due to the number of reviewers, so we will use the general response to highlight points that appeared in one or more reviews. We will then use individual reviewer responses to address the remaining points.

**Problem formulation.** Two reviewers expressed confusion about the problem we address, which we previously referred to as “online feature selection” (oRCJ, 84q8). Our problem formulation is detailed early in the paper (see the abstract, the first paragraph of the introduction, Section 2.2), but it seems that the term “online” caused some confusion. Our choice of terminology was motivated by prior work (Kachuee et al., 2019), but after consulting the literature we have found a better name: “active feature acquisition.” We now use this name throughout the paper, and we also replaced all instances of “offline” feature selection with “static” feature selection. We hope this clarifies any confusion around the terms “online,” “adaptive” and “sequential” (e.g., our problem is unrelated to shifting data distributions, and distinct from standard/static supervised feature selection).

**Novelty.** A couple reviewers raised concerns about novelty, and we can address these points:

- **CMI methods:** we followed up on reviewer 84q8’s comment that CMI is reasonably well known in the feature selection literature, and we found that this is correct to some extent. Several methods explore a similar high-level approach, but crucially, these works lack a reliable implementation. For example, Fleuret (2004) assumes binary features and can only condition on one available feature, and one other work (Ma et al., 2018) requires modeling the conditional distributions of unavailable features using generative models, which is unlikely to work well in practice and is poorly suited to mixed continuous/categorical data. We have updated the paper accordingly by highlighting our main contribution: our implementation based on amortized optimization. Note that Section 3 remains an important part of the paper because it establishes preliminary results that enable our amortized optimization approach; we have clarified this point in our revisions.
- **CAE/STG:** reviewer H6xQ suggested that our method simply replaces the selection layer in STG with the selection layer in CAE. This is a major misunderstanding. The CAE and STG methods both involve learned, unconditional selection layers that are responsible for picking *multiple* features. Our method involves a learned selection *network* that conditions on subsets of available features, and it’s responsible for picking just *one feature* at a time. The mechanism is therefore quite different, and furthermore, an important part of our method is determining the right way to train the selection and prediction networks; this requires more attention in our context than for STG/CAE.

**Related work.** Several reviewers pointed us to specific works, and we’ve improved our related work section accordingly. We’ve made sure to highlight not only these works, but those that were mentioned in their related work, etc.

---

> ### Author Response · Authors · 2022-11-19
> **General response (2/3)**
>
> **Role of RL.** One reviewer misunderstood our method as using RL (84q8). We attempted to make it clear throughout the paper (see the introduction, Section 2.2, Section 4.2) that our method is an alternative to RL. We make this distinction because RL typically optimizes for rewards over multi-step processes, whereas our approach only aims to make a sequence of locally optimal decisions. RL is in general better suited for sequential decision-making problems, but it is often difficult to optimize, so an important finding for our paper is that a greedy algorithm can work better in practice (see our comparison with Kachuee et al., 2019).
>
> **Training objective.** Several reviewers had questions about our training approach and objective function. Briefly, there are no fundamental problems with our approach (its global optimizer is indeed a policy that selects features with maximum CMI), but we were able to improve our presentation in several places. We can address the main points that were raised:
>
> - Reviewer h13P asked why our objective takes an expectation over $p(s)$. This is because it should be possible to make greedy selections starting from any feature set $x_s$, and taking the expectation over all subsets is necessary to learn such a policy. The reviewer is correct that in practice we are most interested in subsets that are likely to be produced by the policy, and that’s exactly why we deviate from this approach in Algorithm 1. So in short, the expectation over all subsets is chosen primarily because it’s theoretically cleaner.
> - Reviewer oRCJ asked what mild assumption we make for Theorem 1. It is simply that there is a unique feature index with maximum conditional mutual information. This is now stated in the text (“we can show the following result under a mild assumption that there is a unique optimal selection”) and also shown in the proof (Appendix A). This also relates to the point raised by h13p: if we relax this assumption, the optimal policy is simply indifferent between the multiple equivalent choices. We have added this explanation to Section 4.1: “If we relax the assumption of a unique optimal selection, the optimal policy is simply indifferent among the optimal indices.”
> - Reviewers oRCJ and dhvL asked other questions about our objective (eq. 8), for example whether $p(s)$ depends on $\theta$ or $\phi$. As we mention in the bullet points at the end of Section 4.2, a key difference between eq. 8 and Algorithm 1 is that we sample subsets according to the learned policy (rather than sampling from $p(s)$), so there is some dependency on $\phi$. However, crucially, gradients are not propagated backwards through the process that yields $x_s$ prior to the selection of the next index. Because of this, the policy and predictor still optimize for one-step performance improvement, and their optimal behavior is unchanged.
>
> This last point is an important observation that needed to be clearer in the text, so we have added an explanatory paragraph to Appendix D immediately following Algorithm 1. We believe this should clarify for readers why it is harmless to generate feature subsets according to the policy in practice (in the sense that it preserves the global optimum).
>
> **Text reorganization.** Multiple reviewers had requests about how to reallocate space in the paper. We were able to make a couple changes:
>
> - Reviewers 84q8 and oRCJ requested that we shorten Section 3 and/or clarify how it is important for developing our main approach in Section 4. Given our increased focus on the implementation via amortized optimization, we have highlighted that Section 3 serves the purpose of deriving preliminary results that help us formulate our learning approach in Section 4. For example, we have added the following text: “Our objective thus optimizes for individual selections and predictions rather than the entire trajectory, which lets us build on earlier results from Section 3.”
> - Reviewer H6xQ asked that we move unnecessary content from Section 4.2 to the supplement. We were able to shorten this section slightly. However, given that there are notable differences between our approach and the CAE/STG methods, it is impossible to remove more content without leaving readers confused (note that multiple reviewers actually requested more information about our training approach).

---

> > ### Author Response · Authors · 2022-11-19
> > **General response (3/3)**
> >
> > **Notation.** Several reviewers had suggestions about how to improve our notation, and we have made updates that should improve readability. First, we removed our set complement notation and $\mathcal{X}, \mathcal{Y}, \mathcal{X}_i$ in Section 2.1 because these weren’t essential (d6k9, H6xQ). Next, we specified that we generally have $k < d$ (H6xQ). We also edited the first sentence of Section 2.1 so that the policy $\pi(x_s)$ is explained immediately after the notation is first used (it was previously explained in the next sentence), and we clarified that minimizing eq. 1 is equivalent to maximizing our final predictive accuracy (H6xQ). We tried replacing our subset notation with a capital $S$ (84q8, oRCJ) but it looks quite bad in subscripts, so we stuck with the original notation $s$. However, we ensured that sets of indices are only ever referred to as $s$, which should minimize reader confusion.
> >
> > **Table 1 results.** Two reviewers thought our results showing the mean AUROC across different numbers of features should be replaced by curves showing the AUROC for each number (H6xQ, 84q8). These curves are already in the paper: Figure 2 shows the curves for our three emergency medicine datasets, and the curves for the remaining tabular datasets are provided in the appendix (“We again applied each method with various feature numbers, and Table 5 shows plots of the AUROC for each budget”). We included Table 1 in the main text to save space and to summarize the consistently strong performance across all six datasets.
> >
> > **Confidence intervals.** Multiple reviewers asked that we include standard deviations for our results Table 1. Thanks for the suggestion, we agree that this is a good idea. We previously presented the best results across multiple trials, but we are working on updating our results to verify that our improvements are statistically significant. Our updated paper contains most of the confidence intervals in Table 1, and we will share the remaining results as soon as possible.

---

### Author Response · Authors · 2022-11-21
**Updated Table 1 results**

To follow up on one of the topics in our response, here are the Table 1 results updated with confidence intervals. The table now shows the mean and standard deviation across five independent trials, and we have the results for all methods except the iterative baseline. (That method is much slower than the others at inference time because it requires making iterating through each remaining feature and making predictions with many possible imputations.) We'll provide the results for that baseline as soon as we can.

But in the meantime, it's apparent from the remaining results that our greedy method provides a statistically significant improvement over the other methods.

|                     | Spam   | MiniBooNE   | Diabetes   | Bleeding   | Respiratory   | Fluid   |
|:--------------------|:-----------------------------|:-----------------------------|:-------------------------------|:-----------------------------|:-----------------------------|:-------------------------------|
| IntGrad      | 88.08 ± 0.36 | 86.77 ± 0.86 | 88.99 ± 0.34 | 69.27 ± 1.18 | 80.42 ± 0.18 | 78.31 ± 0.82 |
| DeepLift      | 89.02 ± 1.51 | 86.33 ± 0.20 | 94.89 ± 0.21 | 67.52 ± 0.27 | 77.66 ± 0.31 | 77.37 ± 0.15 |
| SAGE      | 90.45 ± 0.65 | 91.57 ± 0.10 | 95.46 ± 0.03 | 70.31 ± 0.38 | 82.20 ± 0.37 | 79.60 ± 0.58 |
| Perm Test      | 89.83 ± 0.34 | 92.18 ± 0.15 | 95.50 ± 0.02 | 68.29 ± 0.88 | 80.98 ± 0.27 | 79.24 ± 0.71 |
| CAE      | 92.46 ± 0.29 | 93.04 ± 0.16 | 95.88 ± 0.11 | 70.03 ± 0.37 | 82.78 ± 0.33  | 76.39 ± 1.60 |
| Iterative      | 87.00 ± 1.40 | 92.17 | 95.61 | 70.63 | 78.85 | 84.33 |
| Opportunistic (OL)      | 85.93 ± 0.22 | 69.22 ± 0.73 | 83.06 ± 0.93 | 60.60 ± 0.62 | 74.44 ± 0.48 | 78.12 ± 0.35 |
| Greedy active (ours)      | **94.22 ± 0.20** | **94.23 ± 0.17** | **96.05 ± 0.03** | **72.83 ± 0.31** | **84.36 ± 0.22** | **86.00 ± 0.31** |

Please let us know if you have other questions about our response.

---

### Author Response · Authors · 2022-11-29
**Author response**

Thanks again for the work from all six reviewers to provide feedback on our paper. We worked hard to address the many points you raised, and we want to make sure we can answer any remaining questions/concerns you may have. So we would appreciate it if you could read our response and update your reviews.

We obviously hope that you will be satisfied and improve your scores, but if not, we would still be curious what you expect from an improved version of this work. Everyone seems to agree that the proposed method works quite well, so if there are any remaining presentation concerns we'd be happy to hear what they are.

Thanks,

The authors

---

### Decision · Program_Chairs · 2023-01-20

**Decision:**

Reject

**Justification For Why Not Higher Score:**

I would recommend rejection for this paper based on the following reasons:

- The paper lacks strong theoretical justification and does not adequately cite and compare relevant literature, including works on gradient-based feature selection and weak submodularity. Since submodularity is one of the most important methods in feature selection, it is reasonable to include discussions and comparisons to submodularity, especially in experiments.
- There are questions about the clarity and validity of the main algorithm and Theorem 1 which were not addressed adequately in the authors' response.

**Justification For Why Not Lower Score:**

N/A

**Metareview: Summary, Strengths And Weaknesses:**

Summary
---
The paper presents a method for online feature selection, which is different from the standard feature selection problem where the feature subset is fixed. The proposed method is a greedy strategy that selects features based on their conditional mutual information with the response variable, and uses a deep learning approach to learn the greedy policy. The authors demonstrate through empirical experiments that the proposed method performs well compared to other existing feature selection methods.

Strengths
---
- The proposed method addresses an important and practical problem, namely online feature selection, which has numerous potential applications. The solution is interesting. The writing is clear and easy to read.
- The empirical evaluation is comprehensive, with experiments conducted on multiple datasets and comparisons made against multiple existing methods. The authors also provide their codes for the experiments.

Weaknesses
---
- The paper lacks strong theoretical justification and does not adequately cite relevant literature, including works on gradient-based feature selection and weak submodularity. The section on greedy suboptimality may also be superseded by weak submodularity papers. The authors’ response argued that submodularity doesn’t apply because feasible solutions are represented by policies rather than subsets. However, submodularity is widely used in designing feature selection algorithms and establishing their theoretical guarantees. Even if submodularity may belong to a different family of approaches, since it is one of the most important methods in feature selection, it is reasonable to include discussions and comparisons to submodularity, especially in experiments.
- There are questions about the clarity and validity of the main algorithm and Theorem 1.
- The authors' approach of modeling p(x_i | x_s) is not well-justified and is later bypassed.
- The use of E_s in the cost function as part of the greedy strategy is questionable. The authors’ response did not address this concern adequately.